# Cellular transitions during cranial suture establishment in zebrafish

D'Juan T. Farmer [1] ✉, Jennifer E. Dukov[1], Hung-Jhen Chen [1], Claire Arata [2], Jose Hernandez-Trejo[2], Pengfei Xu[3], Camilla S. Teng[4], Robert E. Maxson[5] & J. Gage Crump [2] ✉

Cranial sutures separate neighboring skull bones and are sites of bone growth. A key question is how osteogenic activity is controlled to promote bone growth while preventing aberrant bone fusions during skull expansion. Using single-cell transcriptomics, lineage tracing, and mutant analysis in zebrafish, we uncover key developmental transitions regulating bone formation at sutures during skull expansion. In particular, we identify a subpopulation of mesenchyme cells in the mid-suture region that upregulate a suite of genes including BMP antagonists (e.g. *grem1a*) and pro-angiogenic factors. Lineage tracing with *grem1a*:nlsEOS reveals that this mid-suture subpopulation is largely non-osteogenic. Moreover, combinatorial mutation of BMP antagonists enriched in this mid-suture subpopulation results in increased BMP signaling in the suture, misregulated bone formation, and abnormal suture morphology. These data reveal establishment of a non-osteogenic mesenchyme population in the mid-suture region that restricts bone formation through local BMP antagonism, thus ensuring proper suture morphology.

The calvarium is the bony shield that covers and protects the brain. In humans, it is composed of the occipital and paired frontal and parietal bones, connected by fibrous joints called sutures that provide flexibility during parturition. Although the types and embryonic origins of the skull bones differ across vertebrates, mice and zebrafish share many of the same sutures with humans and have been relevant models for suture loss in a human birth defect called craniosynostosis[1,2]. The skull bones arise from mesenchymal cells of neural crest or mesoderm origin that condense and grow apically and laterally to cover the brain[3,4]. As bones meet, complex sutures form that integrate osteogenic and connective tissue cells between the overlapping skull bones[5]. Lineage tracing studies in mouse, for example based on genetic recombination mediated by *Gli1*:CreER[6], *Axin2*:CreER[7], and *Prrx1*:CreER[8], suggest that postnatal sutures house osteogenic stem cells that grow the calvaria throughout adulthood. Osteogenic cells at cranial sutures can also be labeled by *Ctsk*:Cre, which labels dedicated stem cells for intramembranous ossification in the periosteum of long bones, suggesting commonalities of bone-forming cells in the calvarium and long bones[9]. Signaling between Ctsk-derived stem cells and a recently identified cartilage-promoting *Ddr2*+ stem cell at cranial sutures further demonstrates the complex crosstalk necessary to maintain a patent suture[10]. Further, suture-residing skeletal stem cells in mice and humans have been captured using surface antigens associated with stem cell identity[9,11,12]. However, the diversity of cells within the suture mesenchyme, and whether subpopulations have differing roles in generating osteoblasts versus regulating the extent of bone formation, remain less defined.

The zebrafish has emerged as a powerful model to interrogate the location and regulation of the progenitors that build and maintain skull bones and sutures. In contrast to mice where key developmental

[1]Department of Molecular, Cell and Developmental Biology, University of California, Los Angeles, CA 90095, USA. [2]Eli and Edythe Broad Center for Regenerative Medicine, Department of Stem Cell Biology and Regenerative Medicine, Keck School of Medicine, University of Southern California, Los Angeles, CA 90033, USA. [3]Department of Orofacial Sciences and Program in Craniofacial Biology, University of California, San Francisco, San Francisco, CA, USA. [4]Department of Cell and Tissue Biology, University of California San Francisco, San Francisco, CA 94143, USA. [5]Department of Biochemistry, Keck School of Medicine, University of Southern California, Los Angeles, CA 90033, USA. ✉e-mail: djuanfar@mcdb.ucla.edu; gcrump@usc.edu

transitions of skull development occur in utero, zebrafish develop outside the mother and are largely transparent. These attributes allow for direct observation of suture formation at cellular resolution in living animals. Recently, we generated a zebrafish model for Saethre-Chotzen syndrome[13], a human birth defect in which the coronal suture is selectively lost due to heterozygous mutations in either *TWIST1* or *TCF12*[14–16]. Homozygous *twist1b; tcf12* mutant zebrafish replicate the specific loss of the coronal suture, which correlates with aberrant skull bone growth in both the fish model and an analogous *Twist1/+; Tcf12/+* mouse model[13]. The extent to which misregulated bone growth reflects cell-autonomous defects in osteoblast lineage cells and/or signaling defects in the suture niche remains an open question, in part due to our incomplete knowledge of the diversity of mesenchyme populations in the suture.

Here, we performed single-cell transcriptomics of the zebrafish skull before and after suture formation, which allowed us to identify distinct mesenchymal populations that build the suture, how they differ before and after suture formation, and how they may be disrupted in craniosynostosis. We identified shared cell types with mouse sutures[5,17], including meningeal subtypes and putative osteoblast progenitors. We were also able to resolve a transcriptionally distinct mesenchyme population located in the mid-suture region that upregulates expression of a number of BMP antagonists (e.g. *grem1a, nog2, nog3*), as well as members of the Angiopoietin family linked to blood vessel development[18]. Whereas an osteoblast-specific nuclear EOS (nlsEOS) photoconvertible transgenic line revealed bone addition primarily at suture edges, short-term lineage tracing with a *grem1a*:nlsEOS reporter revealed *grem1a+* mid-suture cells to be largely non-osteogenic. This mid-suture subpopulation was also enriched for *twist1b* and *tcf12* expression and was greatly reduced in *twist1b; tcf12* mutants that have misregulated osteogenesis. Moreover, combinatorial loss *grem1a, nog2,* and *nog3*, BMP antagonists enriched within the mid-suture mesenchyme subpopulation, resulted in suture-specific upregulation of BMP signaling, misregulated bone formation, and altered suture morphology. These results are consistent with a role of *grem1a+* mid-suture mesenchyme cells in compartmentalizing bone formation to suture edges, thus helping to prevent bone fusion across the body of the suture.

## Results

### Single-cell atlas of calvarial cell types before and after suture formation

To catalog cell types during the transition from growing bone fronts to sutures, we performed single-cell RNA sequencing of dissected zebrafish calvaria at juvenile bone front stages (9–11 mm standard length (SL)) and adult stages when sutures are fully established (22–25 mm SL) (Fig. 1A). After filtering with Seurat v4.1[19], we recovered 16,245 cells (median of 719 genes per cell) at juvenile stages and 5066 cells (median of 683 genes per cell) at adult stages. Unsupervised clustering of the integrated datasets identified 16 clusters (Fig. 1B). We identified chondrocytes, osteoblasts, and mesenchyme clusters (including dermal fibroblasts defined by high expression of Phe/Tyr metabolism genes[20]), as well as endothelial cells, epithelial cells, glial cells, hair cells from neuromasts, immune cells, red blood cells, and neuronal cell types (Fig. 1B, Supplementary Fig. 1A, Supplementary Data 1). The immune population, which we do not further investigate in this study, includes cells expressing markers for T-cells, macrophages, and neutrophils, and a cluster with a mixed signature for natural killer cells and T-cells[21] (Supplementary Fig. 1A). We noted differences in the relative abundance of cell types between stages, particularly the enrichment of the mesenchyme cluster at adult stages (61% of all captured cells) and neuronal and immune subtypes at juvenile stages, although the extent to which this reflects relevant biological versus stage-dependent technical differences remains unclear (Supplementary Fig. 1B, C).

To better understand the mesenchymal and osteogenic cell types captured from our datasets, we isolated the connective tissue and skeletogenic clusters (Fig. 1B, dotted line) and repeated unsupervised clustering of the integrated subset to uncover 18 clusters (Fig. 1C, Supplementary Data 2). As we validated that *prrx1a* expression broadly labels mesenchyme within and around adult cranial sutures[13] and *pah* expression labels dermal fibroblasts outside of sutures (Supplementary Fig. 2A, B), we created a new subset that included the *ifitm5+* osteoblast cluster and *prrx1a+* mesenchymal clusters, after removing *col2a1a+* chondrocytes, *pah+* dermal fibroblasts, and *acta2+; csrp1b+* vascular smooth muscle cells[22] (Fig. 1C, Supplementary Fig. 2C). After re-clustering, we obtained 8 clusters (517 juvenile cells, 656 adult cells) (Fig. 1D). Analysis of enriched genes for each cluster (see Supplementary Data 3) allowed us to assign preliminary cell identities (Fig. 1D, E), which we validate in this study. Based on known marker genes, these clusters include two meningeal cell types, osteogenic mesenchyme, periosteum, pre-osteoblasts, and early and late osteoblasts, as well as a population we validate as largely mid-suture mesenchyme. All clusters are composed of cells from juvenile and adult stages (Supplementary Fig. 2D) with some differences in relative composition (Supplementary Fig. 2E), such as increased osteoblasts and decreased osteogenic mesenchyme at adult stages that likely reflects slowing of bone growth. We also performed RNA velocity to infer the directionality of cell state changes based on nascent gene expression. Osteogenic mesenchyme flows toward the committed osteoblast subtypes (pre-osteoblast, early and late osteoblasts), and mid-suture mesenchyme has weak connections toward the periosteum and osteogenic mesenchyme (Fig. 1F). This analysis supports osteogenic and not mid-suture mesenchyme being the main precursor to new osteoblasts in the calvarium, although the weak flow of regulatory to osteogenic mesenchyme might suggest that the mid-suture mesenchyme has a latent ability to contribute to the osteogenic lineage.

### Cell type conservation between zebrafish and mouse sutures

Whereas zebrafish, mice, and humans share genetic requirements for cranial suture formation[13], the similarities of suture-related cell types across vertebrates was unclear. To address this, we compared cell clusters in our zebrafish skull datasets to those identified from the late embryonic mouse coronal and frontal sutures[5,17]. Mapping of homologs of genes enriched from each mouse cell type onto our zebrafish dataset revealed a number of cell types in common (Fig. 1G, H). The meninges have been implicated in cranial suture patency and regeneration[23–26] and are composed of at least three layers: the pia mater found closest to the brain, the arachnoid mater, and the dura mater that underlies the calvarium. Homologous genes for the periosteal dura cluster MG2 from the mouse coronal suture dataset showed enriched expression in zebrafish cluster periosteal dura, and homolog expression for mouse dura mater cluster MG3 and arachnoid mater cluster MG4 was enriched in zebrafish cluster meningeal-other (Fig. 1G). The zebrafish cluster meningeal-other also shared gene expression with the meningeal cluster DM from the mouse frontal suture dataset (Fig. 1H). In situ RNA hybridization for the meninges-other enriched marker *zic3*, and the periosteal dura marker *foxc1b* with the pan-meninges marker *crhbp*, identified double-positive cells concentrated beneath calvarial bones in adult zebrafish (Supplementary Fig. 3A, B). For the ectocranium, homologs of mouse EC1-3 markers (*chl1a, tek, nefl*) were co-enriched in the *pah+* zebrafish dermal fibroblast cluster 14, and markers for frontal suture cluster FS1 were apparent more broadly within *prrx1a+* mesenchyme (Supplementary Fig. 4A, B). Frontal suture cluster FS2 showed homology to the zebrafish vascular smooth muscle cell cluster 12. Coronal suture ectocranium layer EC4 and ligament layer LIG, and frontal suture hypodermis cluster HD, were not clearly homologous to any clusters in our zebrafish dataset (Supplementary Fig. 4A, B).

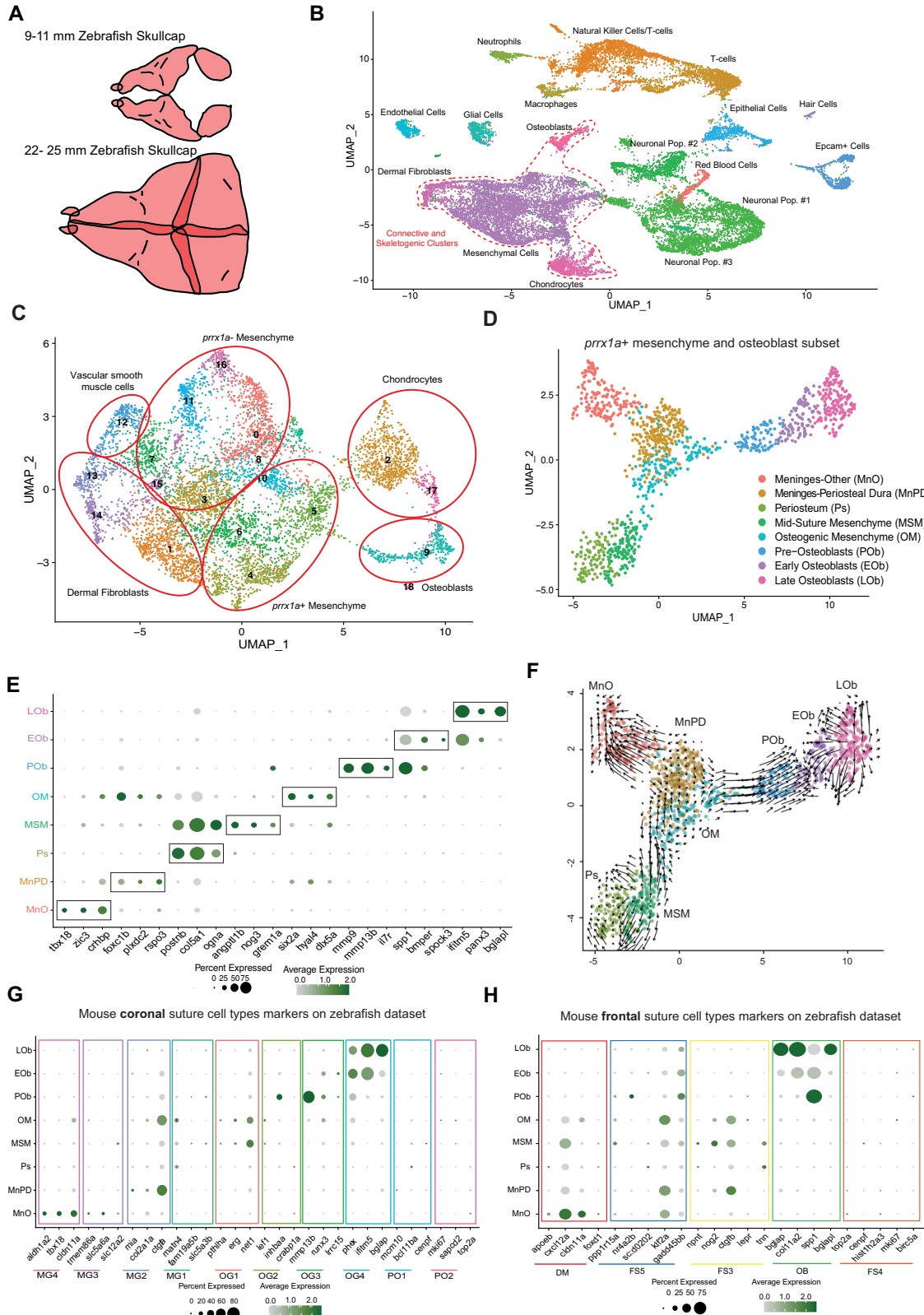

**Fig. 1 | Single-cell RNA-sequencing analysis of juvenile and adult calvaria.**
**A** Schematic of dissected calvaria from juvenile and adult zebrafish. **B** Uniform Manifold Approximation and Projection (UMAP) plot of integrated juvenile (16,245 cells) and adult (5066 cells) datasets. Connective and skeletogenic clusters outlined by dashed lines. **C** UMAP analysis of reclustered connective and skeletogenic subset, including juvenile (4324 cells) and adult (4595 cells) datasets. Cell identities grouped by circles. **D** UMAP plot of osteoblast and *prrx1a*+ mesenchyme subset

(517 juvenile cells, 656 adult cells). **E** Dot plot showing markers enriched for each cluster within the osteogenic/mesenchymal subset. **F** RNA velocity (arrows) showing differentiation flow for cells within the osteogenic/mesenchymal subset. **G**, **H** Dot plots showing zebrafish paralogs for markers of osteogenic and meningeal clusters in the mouse coronal or frontal suture datasets mapped onto the zebrafish osteoblast/mesenchyme subset.

For the osteoblast lineage, mouse osteoprogenitor cluster OG1 from the coronal suture dataset, as well as FS3 from the frontal suture dataset, share gene expression in common within zebrafish clusters mid-suture mesenchyme (MSM) and osteogenic mesenchyme (OM) (Fig. 1G, H). We also identified a zebrafish pre-osteoblast cluster with similarity to mouse pre-osteoblast clusters OG2 and OG3, and two zebrafish clusters with similarity to coronal suture osteoblast cluster OG4 and frontal suture osteoblast cluster OB, which we term early osteoblast and late osteoblast based on differential expression of mature osteoblast markers such as *bglap and bglapl* (Fig. 1G, H). Markers for the proliferative osteogenic cell populations PO1 and PO2, as well as proliferative cluster FS4, were not enriched within any zebrafish cluster; instead, we note broad expression of the proliferative marker *pcna* across clusters, including pre-osteoblasts and osteoblasts (Supplementary Fig. 2F). Additionally, markers for frontal suture cluster FS5, predicted to be an artifact of dissociation, were not enriched within any zebrafish cluster. These data indicate conservation of many meningeal and osteogenic cell types between zebrafish and mouse calvaria.

### Signaling interactions across suture cell types

To interrogate the transcriptional programs enriched within each cell type, we performed GO analysis using the significantly enriched genes for each cluster in our dataset (Supplementary Fig. 5A). OM and osteoblast populations were enriched for terms associated with the skeletal system, cellular respiration, and extracellular matrix organization. MSM displayed enrichment in blood vessel morphogenesis and regulation of BMP signaling pathway, and meninges displayed enrichment for BMP, Wnt, and retinoic acid signaling pathways. Periosteal dura is enriched for cartilage development, consistent with the expression of cartilage-associated markers in its homologous population in the mouse coronal suture[5]. We next performed CellChat analysis[27] to assess potential signaling interactions between cell types. Whereas OM and cartilage-related periosteal dura were predicted to have largely incoming signaling, MSM and late osteoblasts have largely outgoing signaling (Supplementary Fig. 5B). Moreover, interaction mapping suggests the strongest predictive interactions arise from both MSM and late osteoblasts toward periosteal dura and OM (Supplementary Fig. 5C, Supplementary Data 4). These data suggest that signaling from MSM, and likely also negative feedback from differentiated osteoblasts, regulates OM and periosteal dura populations.

### Shared gene expression between bone front cells and suture mesenchyme

As the developmental origin and composition of the mesenchyme that supports calvarial expansion prior to suture formation is poorly understood, we first sought to discern the spatial architecture of mesenchyme associated with the growing skull bones. Labeling of the neural crest cells that give rise to the anterior portion of the frontal bone, either by blastula-stage transplantation of GFP+ neural crest precursors or mosaic *Sox10*:Cre-mediated neural crest-specific DNA recombination of a fluorescent reporter, revealed a layer of mesenchyme cells ahead of *RUNX2*:GFP+ pre-osteoblasts and Alizarin Red-stained bone at juvenile stages (10 mm SL) (Fig. 2A, B). In order to examine whether the mesenchyme ahead of the osteoblast layer corresponds to the putative mesenchyme clusters identified in our single-cell datasets, we examined expression of genes enriched in the OM and MSM clusters (Fig. 2C). We focused on *six2a* and the ETS factor *fli1a*, as we had previously found *Six2* and the related ETS factor *Erg* to be markers of osteoblast progenitors at the mouse embryonic coronal suture[5] (Fig. 1F, G). In situ mRNA hybridization confirmed *six2a* expression in the mesenchyme layer ahead of the growing juvenile frontal bone, and at both bone tips and mid-suture regions of the coronal suture at adult stages (Fig. 2D, E). Similar patterns of *fli1a* expression were seen with a *fli1a*:GFP reporter, in combination with

Calcein Blue staining of bone or a *sp7*:mCherry osteoblast reporter (Fig. 2F–I). Although additional lineage tracing experiments will be required, these results suggest that suture mesenchyme arises, at least in part, from mesenchyme at the leading edges of the growing calvarial bones prior to suture establishment.

### Osteoblast addition is largely restricted to suture edges

Establishment of sutures coincides with a transition from rapid growth of the embryonic calvarial bones to more limited growth that expands the skull while preventing inappropriate bony fusions[28]. To understand this transition, we first characterized the trajectory of osteoblast differentiation in the zebrafish calvaria. Using known marker genes as a guide, we identified clusters corresponding to presumptive pre-osteoblasts (*mmp9+*, *mmp13b+*, *spp1+*, *runx2b+*), early osteoblasts (*spp1+*, *ifitm5+*, *sp7+*, *bglapl-*), and late osteoblasts (*spp1+*, *ifitm5+*, *sp7+*, *bglapl+*) (Fig. 1E, Supplementary Fig. 6). Although *Mmp9* has not been described as a marker for pre-osteoblasts during mouse intramembranous ossification, *mmp9* has been shown to mark pre-osteoblasts during regeneration of intramembranous bone in the zebrafish fin[29]. In addition, *Mmp9* has been shown to have redundant functions with *Mmp13* in mouse long bone development[30], with *Mmp13* also marking pre-osteoblasts in the mouse coronal suture[5]. Here we show that *mmp9* and *mmp13b* are co-expressed in pre-osteoblasts of the zebrafish calvarium. To interrogate gene expression dynamics along osteogenic differentiation, we performed Monocle3 analysis. We used OM as the root based on its shared gene expression program with osteoprogenitors from mouse and zebrafish scRNA-seq datasets[5,20], and the transcriptional flow from OM to committed osteoblast identities in our RNA velocity analysis. Consistent with RNA velocity analysis (Fig. 1F), pseudotime analysis using Monocle3 predicts a trajectory from the *six2a* + OM cluster to *mmp9+*, *spp1+* pre-osteoblasts; *spp1+*, *ifitm5+* early osteoblasts; and *bglap1+* late osteoblasts (Fig. 3A-C). This suite of markers is shared between juvenile and adult stages, suggesting similar trajectories of osteoblast differentiation (Fig. 3D).

To discern the spatial architecture of osteogenic cell types, we used a combination of transgenic labeling and mRNA in situ hybridization. In particular, we made use of the finding that *mmp9* expression is highly specific for the pre-osteoblast cluster, with little to no expression in differentiated osteoblasts (Fig. 3C, D). Prior to suture formation *mmp9* expression was highly localized to the growing tip of the frontal bone, and after suture formation remained at the tips of the frontal and parietal bones and was generally excluded from the mid-suture region (Fig. 3E, F). We confirmed localized expression of *mmp9* to suture edges using an *mmp9*:GFP transgenic line that has been shown to mark osteoblast progenitors in the regenerating fin[29] (Fig. 3G). In contrast, expression of osteoblast markers *spp1*, *ifitm5*, and *sp7*:mCherry was observed more broadly along bone surfaces at juvenile and adult stages, including in osteoblasts lining bone surfaces within sutures (Fig. 3E–G). These findings suggest that bone formation is largely restricted to growing bone tips and adult suture edges in zebrafish.

To confirm that osteoblasts are preferentially generated at suture edges, we used a recently created osteoblast-enriched nlsEOS reporter identified while screening for knock-in lines at the *angptl1b* locus. Co-localization with previously characterized *sp7*:mCherry and *RUNX2*:GFP transgenic lines confirmed osteoblast enrichment of the *osteoblast*:nlsEOS line, though expression was also seen in the skin, chondrocytes, and other tissues outside the skull (Fig. 4A, Supplementary Fig. 7A–C). Exposure to UV light converts the green nlsEOS fluorescent protein to red (schematized in Fig. 4B, E). Following conversion of *osteoblast*:nlsEOS to red fluorescence at juvenile bone front stages, new (i.e. unconverted green) osteoblasts appeared at bone fronts within 24 hours (Fig. 4C, D). Previously converted (i.e. red) osteoblasts also continued to produce new unconverted (i.e. green) nlsEOS, seen as white in the merged channel, with digital subtraction of

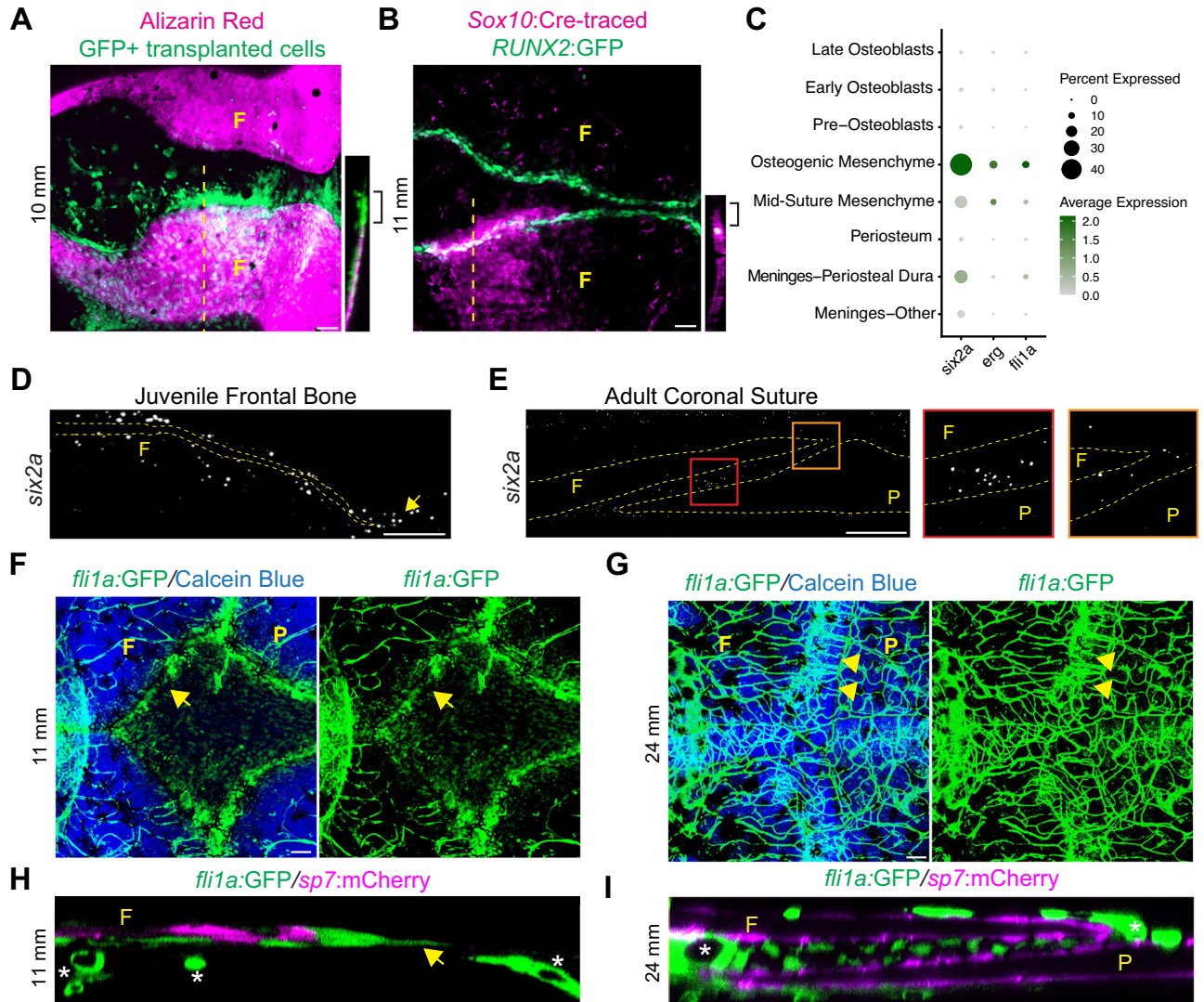

**Fig. 2 | Conserved features of juvenile bone front and adult suture mesenchyme. A** Confocal image of juvenile frontal bones (Alizarin Red+) from animals receiving GFP+ neural crest precursor transplants. Dotted line indicates region for adjacent orthogonal projection. Brackets denote mesenchyme ahead of the growing frontal (F) bone. $N = 3$. **B** Confocal image of juvenile frontal bones. Mosaic recombination within *Sox10*:Cre; *actab2*:loxP-BFP-STOP-loxP-dsRed fish labels a neural crest clone in magenta, and *RUNX2*:GFP labels nascent osteoblasts. Dotted line indicates region for adjacent orthogonal projection. Brackets denote mesenchyme ahead of the bone front. $N = 5$. **C** Dot plot of mesenchyme-enriched genes. **D**, **E** In situ hybridization for *six2a* in the juvenile frontal bone and adult coronal suture. Dotted lines outline the frontal (F) and parietal (P) bones. Arrow marks the tip of the growing frontal bone. Orange boxed inset shows expression of *six2a* at suture edges and red boxed inset shows *six2a* expression in mid-suture region. In situs were performed in biological triplicate. **F**, **G** Confocal images show *fli1a*:GFP+ mesenchyme relative to Calcein Blue+ calvarial bones at juvenile and adult stages. Arrow denotes *fli1a*:GFP+ cells ahead of the growing frontal bone, and arrowheads suture mesenchyme expression. Note extensive labeling of vasculature by *fli1a*:GFP. $N = 5$ each stage. **H**, **I** Orthogonal sections of confocal images show *fli1a*:GFP+ cells at the juvenile frontal bone tip (H, arrow) and adult suture relative to *sp7*:mCherry+ osteoblasts. Asterisks mark *fli1a*:GFP+ endothelial cells. $N = 3$ each stage. Scalebars: 50 µM (**A**, **B**), 100 µM (**D**–**I**).

the red from green channel highlighting the preferential localization of new osteoblasts at the bone front. At adult stages, photoconversion of *osteoblast*:nlsEOS followed by a 7-day chase revealed enrichment of new osteoblasts at the edges of the sagittal suture (Fig. 4F, G). These results confirm our marker analysis and previous reports in mice[31] that osteoblast differentiation occurs primarily at the growing bone fronts and then at the edges of the suture.

### Localization of a largely non-osteogenic mesenchyme subtype to the mid-suture region

Whereas in mouse we had identified a single *Erg*+; *Six2*+ population containing cells with osteoprogenitor properties (OG1), the analogous *six2a*+; *erg*+ population from our zebrafish dataset resolved into two distinct clusters, OM and MSM (Fig. 2C, E). The OM cluster is enriched

for the ETS factor *fli1a* and *hyal4*, which we previously showed broadly label osteochondral progenitors of the neural crest lineage[20] (Fig. 5A). In contrast, while sharing expression of multiple genes (e.g. *col5a2, col12a1a, tnmd, ogna, postnb*) with the periosteum cluster, MSM differs from OM and periosteum clusters by expression of several BMP antagonists (*grem1a, nog2, nog3, bambia, fstl3*), the Tgfb antagonist *tgfbi*, and members of the Angiopoeitin-like (Angptl) family (*angptl1a, angptl1b, angptl2b, angptl5*) that have been implicated in regulation of blood vessel development[18,32] (Fig. 5A; Supplementary Fig. 8A). Moreover, several of these genes (e.g. *angptl1b, grem1a, nog2, nog3, angptl5, bambia*) are selectively expressed in MSM cells at adult suture versus juvenile bone front stages (Fig. 5B, Supplementary Fig. 8B). In situ mRNA hybridization revealed that *ogna*, a gene in common between MSM and periosteum, was broadly expressed along the surfaces of the

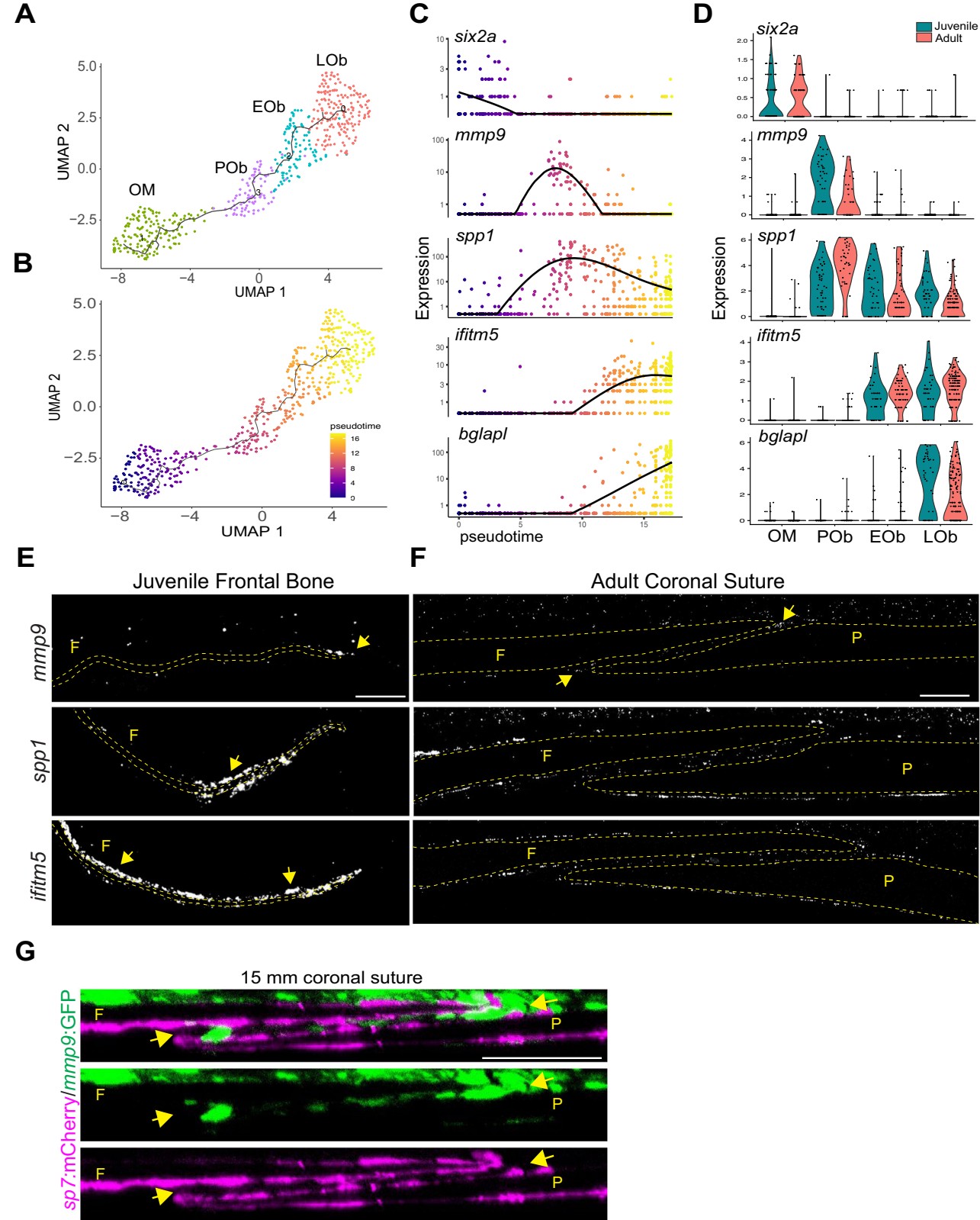

**Fig. 3 | scRNA-seq resolves osteoblast trajectories in the skull. A** Lineage analysis and (**B**) pseudotime analysis of reclustered osteogenic subset using Monocle 3. The black lines in (**A**) represent the lineage relationship between cells along the trajectory. In (**B**), cells at the beginning of the trajectory are darkest (purple) and more mature cells are progressively lighter (orange and yellow). **C** Expression plots of select genes across pseudotime. **D** Violin plot split by age for osteogenic markers from Monocle analysis. **E**, **F** In situ analysis of juvenile frontal (F) bone and adult coronal sutures for *mmp9*, *spp1*, and *ifitm5*. Dotted lines outline the frontal and parietal (P) bones. Arrows mark enriched signal at bone tip and suture edges (*mmp9*) or along bone surfaces (*spp1*, *ifitm5*). In situs were performed in biological triplicate. **G** Orthogonal section of adult coronal suture labeled by *mmp9*:GFP and *sp7*:mCherry. Arrows mark edges of the coronal suture. *N* = 2. Model of osteogenic cell types at bone fronts and cranial sutures. Scalebars: 100 μM.

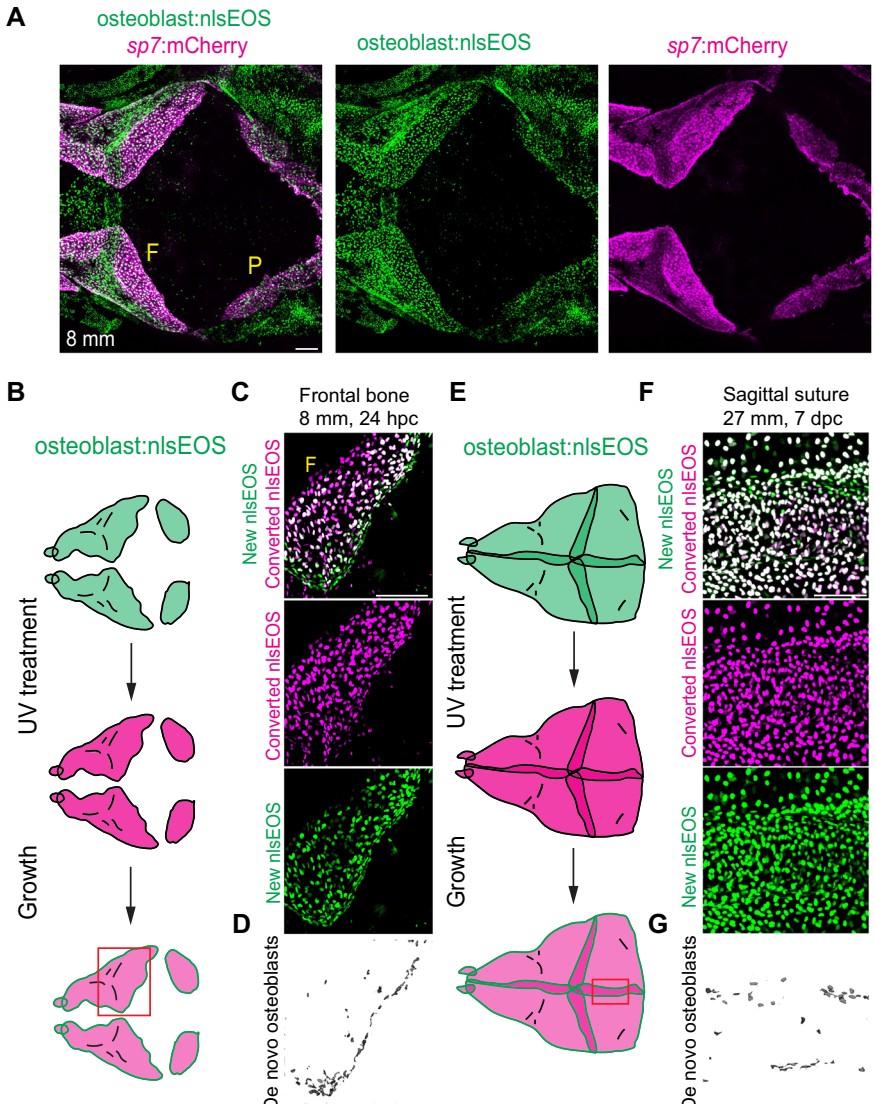

**Fig. 4 | Spatially restricted osteogenesis at bone fronts and suture edges.**
**A** Confocal image of juvenile skull (dorsal view) showing *osteoblast*:nlsEOS relative to the osteoblast marker *sp7*:mCherry in the frontal (F) and parietal (P) bones. *N* = 6. **B** Schematic of treatment scheme for juvenile calvaria. Red box indicates area imaged in the next panels. **C**, **D** Confocal images of *osteoblast*:nlsEOS+ frontal bone 24 h post-conversion (hpc) showing merged and individual channels, and extracted

de novo osteoblasts (green only). *N* = 3. **E** Schematic of treatment scheme for adult calvaria. Red box indicates area imaged in the next panels. **F**, **G** Confocal images of *osteoblast*:nlsEOS+ sagittal suture 7 days post-conversion (dpc) showing merged and individual channels, and extracted de novo osteoblasts (green only). Scalebars: 100 µM (A), 50 µM (**C**, **F**).

growing frontal bone at juvenile stages and the frontal and parietal bones at adult stages, as well as throughout coronal suture mesenchyme (Fig. 5C). In contrast, expression of MSM-specific genes *angptl1b*, *grem1a*, and *nog3* were confined to mid-suture regions at adult stages and largely absent from juvenile bone fronts, with the exception of weak *angptl1b* expression at juvenile bone tips (Fig. 5C). Multicolor in situ hybridization for *angptl1b* with *angptl2b* or *grem1a* confirmed co-expression of these markers within the mid-suture mesenchyme (Supplementary Fig. 8C, D). In contrast, *hyal4* and *fli1a*, markers of the OM population, were largely restricted to the edges of the adult coronal suture (Fig. 5D, E), although we detected some co-expression of *hyal4* with *angptl1b* (Supplementary Fig. 8E). These findings point to spatial segregation of distinct subtypes of *six2a*+ mesenchyme in the adult coronal suture to suture edges and mid-suture mesenchyme.

To test whether an analogous MSM signature might exist in the mid-suture region in mouse, we assessed the distribution of mouse genes homologous to a set of the most restricted MSM

genes in zebrafish (*Angptl2*, *Grem1*, *Nog*, *Tgfbi*), relative to a randomized gene set, across osteogenic subtypes from the E15/E17 coronal suture scRNAseq dataset[5] (Supplementary Fig. 9A). The zebrafish MSM signature mapped most strongly to the mouse OG3 periosteum pre-osteoblast cluster (Supplementary Fig. 9B, C). We also observed significant mapping to the mouse OG1 osteoprogenitor cluster, suggesting that, as in zebrafish, a mouse MSM population might be embedded in a broader osteoprogenitor population and share a transcriptional signature with the periosteum. Combinatorial in situ analysis relative to *Sp7*+ osteoblasts revealed enriched co-expression of *Angptl2*, *Nog*, and *Tgfbi* within the mid-suture region of the mouse coronal suture at postnatal day 2, although some expression was also observed along the periosteum of the frontal and parietal bones (Supplementary Fig. 9D, E). These results suggest the presence of a conserved signature for MSM within the mid-suture region of the mouse coronal suture.

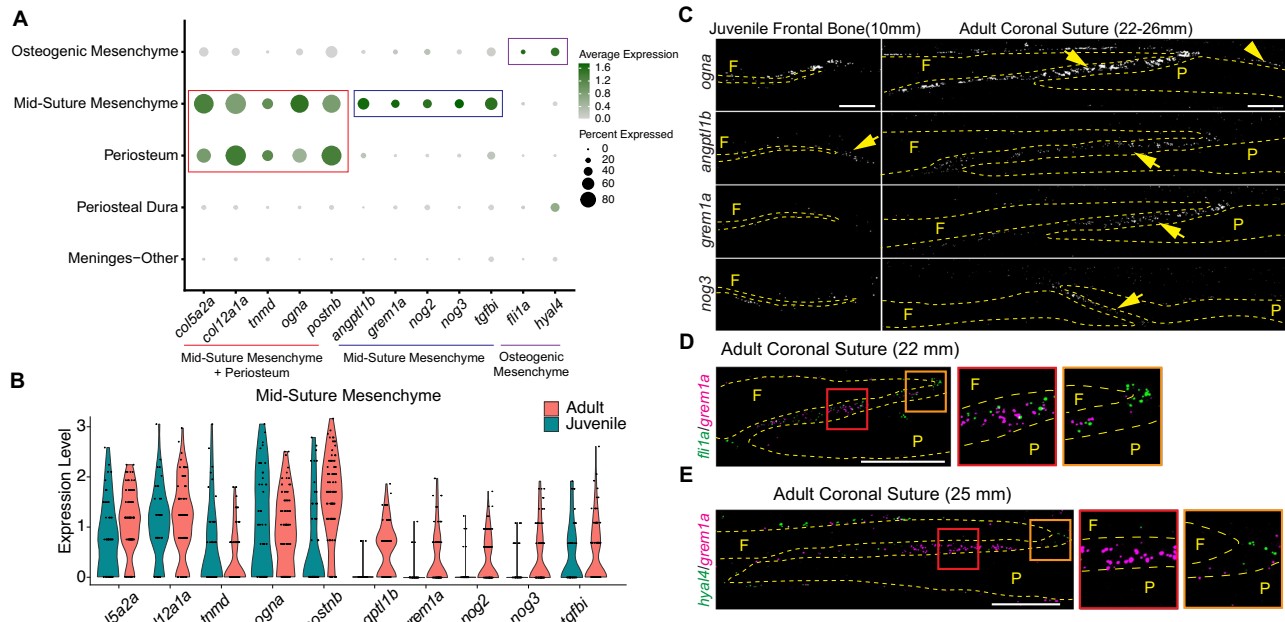

**Fig. 5 | Establishment of distinct molecular signature for mid-suture mesenchyme. A** Dot plot shows shared and cell type-specific markers for subsets of mesenchyme. **B** Violin plot split by age for genes expressed in regulatory mesenchyme. **C** In situ analysis of juvenile frontal bone and adult coronal suture. Dotted lines outline the frontal (F) and parietal (P) bones. Arrows mark bone front and suture mesenchyme, arrowheads the periosteum. **D, E** Double in situ analysis of adult coronal suture. Orange boxed insets show expression of *fli1a* and *hyal4*, but not *grem1a*, at suture edges, and red boxed insets show *grem1a* but not *fli1a* expression in mid-suture region. In situs were performed in biological triplicate. Scalebars: 100 μM.

## grem1a+ mid-suture mesenchyme cells make minimal short-term contributions to osteoblasts

To test the contribution of MSM to new osteoblasts, we performed lineage tracing of these cells with a recently created *grem1a*:nlsEOS knock-in transgenic line. Consistent with expression of endogenous *grem1a* mRNA, *grem1a*:nlsEOS is absent from growing bone fronts and first visible in mid-suture mesenchyme across all sutures as calvarial bones meet (Fig. 6A). To test if nlsEOS can function as a lineage reporter, we tracked photoconverted skulls one month after UV treatment (Fig. 6B, C). Previous observations suggest that nlsEOS protein can perdure for several weeks after expression of nlsEOS mRNA ceases[33,34], and we find that mid-suture mesenchyme retains detectable converted protein and accumulates new nlsEOS protein, dynamics consistent with its use as a lineage reporter (Fig. 6B, C). During this time period, *grem1a*:nlsEOS+ cells remained confined to the mid-suture region and did not noticeably change their distribution. Despite nlsEOS protein stability over several weeks, we observed no co-expression of nlsEOS with the osteoblast marker *sp7*:mCherry, showing that *grem1a*+ cells do not generate osteoblasts within a month (Fig. 6D, E). In contrast, quantification of de novo osteoblast at adult stages (>24 mm) using *osteoblast*:nlsEOS confirmed sustained osteogenesis at all cranial sutures within a two-week period (Supplementary Fig. 10). These results point to *grem1a*+ MSM cells not being a major contributor to new osteoblasts, although we cannot rule out that they make contributions to osteoblasts in non-homeostatic conditions.

## MSM loss and reduced blood vessels in a zebrafish model of Saethre Chotzen Syndrome

We next analyzed how zebrafish suture cell types are affected in *twist1b*[−/−]; *tcf12*[−/−] mutants with coronal suture fusion. In our single-cell data, the highest expression of *twist1b* and *tcf12* is in the MSM population, which we confirm by co-expression of the MSM marker *angptl1b* and *twist1b* in the mid-suture region of the coronal suture (Fig. 7A, B). In *twist1b*[−/−]; *tcf12*[−/−] mutants, we observed a near complete absence of *grem1a*:nlsEOS+ cells in the fused coronal suture region, and a

reduction in the width of the *grem1a*:nlsEOS+ MSM domain in unfused regions of the coronal sutures, and to a lesser degree at the sagittal sutures (Fig. 7C, D). Our *osteoblast*:nlsEOS reporter allows us to quantify osteoblast differentiation and readily visualize the location of cranial sutures by detection of de novo osteoblasts (green) and increased signal from two overlapping bones covered by osteoblasts. Using *osteoblast*:nlsEOS to quantify osteoblast addition, we found that loss of *grem1a*:nlsEOS+ MSM cells in *twist1b*[−/−]; *tcf12*[−/−] mutants correlated with increased osteoblast formation at growing bone fronts (Fig. 7E, F). At suture stages, osteoblast addition has greatly decreased in wild-type controls, with sutures identified by increased signal from the two overlapping bones and presence of new green osteoblasts. In contrast in mutants, we observed a lack of new osteoblast formation in regions of the fused coronal suture, consistent with previous findings based on bone mineralization stains[13]. Using the *fli1a*:GFP reporter, we also noted that blood vessels became concentrated at sutures in wild types (Fig. 2F, G), with mutants displaying a significant reduction in blood vessel density at fused coronal sutures (Fig. 7G, H), consistent with GO term enrichment of blood vessel morphogenesis and expression of the Angptl class of angiogenic proteins in MSM cells. These results demonstrate that reduced numbers of *grem1a* + MSM cells in *twist1b*[−/−]; *tcf12*[−/−] mutants correlate with misregulation of bone formation and a reduction of blood vessels at the coronal suture.

## BMP antagonists are required for proper bone growth and suture morphology

To test whether BMP antagonists, which are preferentially enriched in the MSM population, are required for calvarial development, we generated small deletion alleles for *grem1a*, *nog2*, and *nog3* that result in premature protein truncations (Fig. 8A). Homozygous mutants for *grem1a* or *nog2* but not *nog3* were adult viable. Whereas homozygous *grem1a* mutants formed all cranial sutures based on Alizarin staining of bone (Supplementary Fig. 11A), quantification of osteoblast addition by serial photoconversion of osteoblast:nlsEOS revealed a mild decrease in bone formation in the coronal but not metopic suture

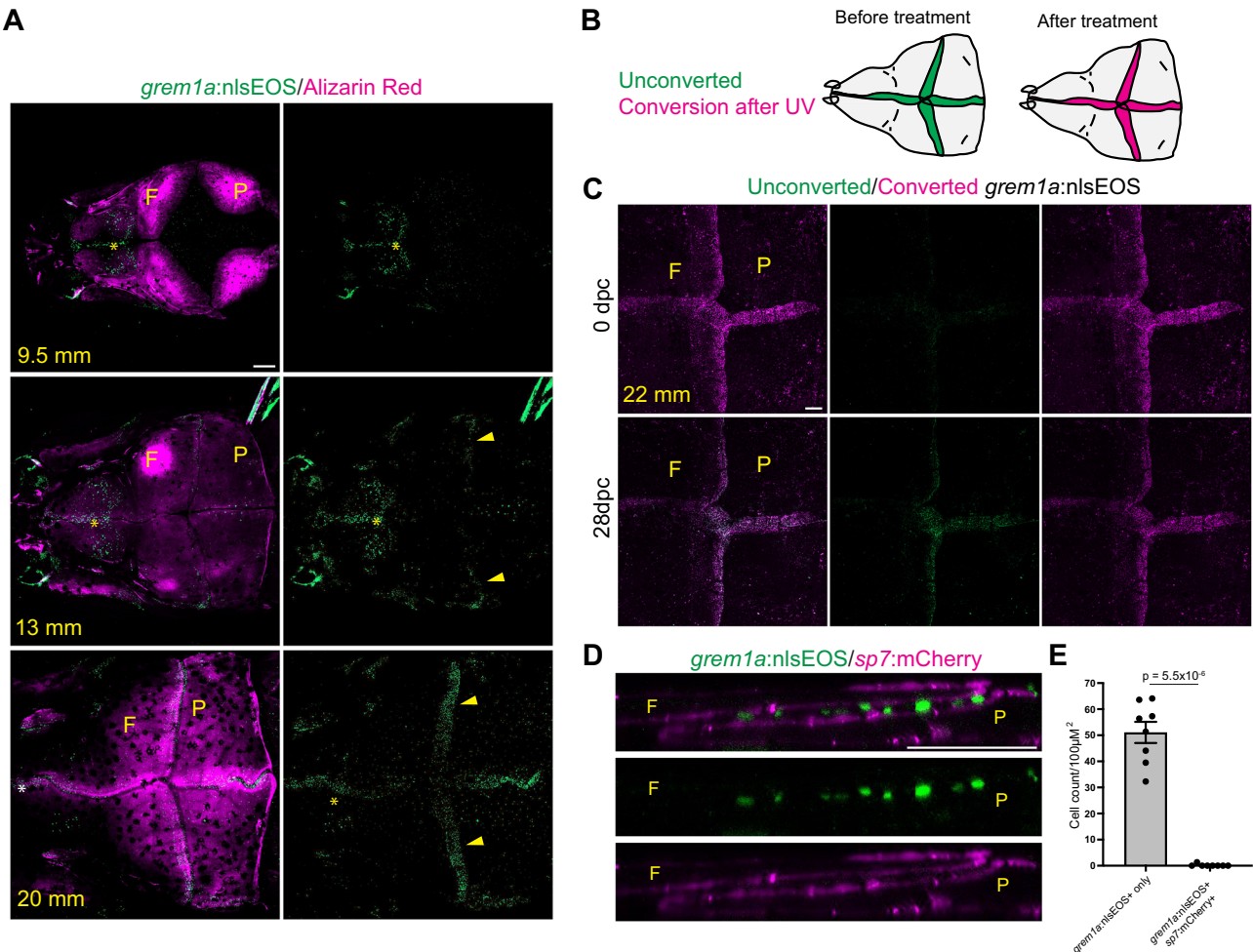

**Fig. 6 | Mid-suture mesenchyme upregulates *grem1a* and does not contribute to bone. A** Confocal images show Alizarin Red staining of frontal (F) and parietal (P) bones relative to induction of *grem1a*:nlsEOS in forming sutures (arrowheads). Asterisks denote brain expression. *N* = 5 each stage. **B** Schematic of conversion scheme for calvaria. *grem1a*+ cells were globally converted, imaged, and re-imaged at 28 days post conversion (dpc). **C** Confocal images of *grem1a*:nlsEOS+ calvaria at 0 and 28 dpc. *N* = 6. **D** Orthogonal section of adult coronal suture shows *grem1a*:nlsEOS in mid-suture cells non-overlapping with *sp7*:mCherry+ osteoblasts. *N* = 8. **E** Quantification of *grem1a*:nlsEOS single-positive and *grem1a*:nlsEOS; *sp7*:mCherry double-positive cells at cranial sutures. Scalebars: 200 μM (**A**, **B**), 50 μM (**D**). *P*-values were calculated using a two-tailed non-parametric Student's *t* tests. Error bars represent S.E.M. Source data are provided as a Source Data file.

(Supplementary Fig. 11B–D), similar to what is observed in *twist1b*; *tcf12* mutants[13]. We observed a significant deceleration of frontal bone growth in *grem1a* mutants between 13 and 15 mm SL stages by serially staining with different color bone dyes as previously reported[13], which may reflect a higher sensitivity of the assay to capture cumulative bone growth changes compared to quantification of de novo osteoblasts during the same growth period (Supplementary Fig. 11E–G). While skull shape was not obviously abnormal in *grem1a* mutants, we did not perform a comprehensive analysis.

Reasoning that BMP antagonists may have redundant functions in regulating calvarial development, we generated compound mutants. Of 659 fish genotyped at 14 days post-fertilization (dpf) from *grem1a*^+/−; *nog2*^+/−; *nog3*^+/− incrosses, we did not recover any triple homozygous mutant fish. We therefore focused on *grem1a*^−/−; *nog2*^−/−; *nog3*^+/− fish, which survive beyond suture formation stages. The *grem1a*^−/−; *nog2*^−/−; *nog3*^+/− mutants displayed highly dysmorphic sutures at 17 mm (Fig. 8B), reflected by an end-on-end rather than overlapping coronal suture and a general appearance of open space between bones at all sutures due to a failure of bone overlap, (Fig. 8C). However, animals die before reaching adulthood for unknown reasons, precluding prolonged analysis to test for eventual craniosynostosis (Fig. 8B). Serial bone dye labeling of *grem1a*^−/−; *nog2*^−/−; *nog3*^+/− mutants revealed

accelerated bone formation between 11–13 mm, when the coronal suture is being established, and a subsequent decrease in bone formation after suture establishment (13–15 mm) (Fig. 8D–F). We also observed increased supernumerary bones at the posterior edge of the sagittal suture in *grem1a*^−/−; *nog2*^−/−; *nog3*^+/− mutants versus wild types and *grem1a* single mutants (Fig. 8G, H), and ectopic bone islands within the coronal suture in more than half of *grem1a*^−/−; *nog2*^−/−; *nog3*^+/− fish but not *grem1a* mutants or wild types (Fig. 8I, J). Quantification of phosphorylated SMAD1/5 (pSMAD1/5), a marker of active BMP signaling, at 14 mm demonstrated an increase in the fraction of positive cells within the coronal suture region, but not along bone surfaces outside the suture, in *grem1a*^−/−; *nog2*^−/−; *nog3*^+/− mutants, consistent with a local increase in BMP signaling at the mutant suture (Fig. 8K, L). These data suggest a requirement for multiple BMP antagonists enriched in the MSM population to restrict bone formation in the calvaria and ensure proper suture morphology.

## Discussion

Here we reveal a high degree of spatial compartmentalization of osteogenesis in cranial sutures that is essential to sustain bone growth while preventing precocious osteoblast differentiation (Fig. 9). By integrating single-cell sequencing and in vivo expression validation

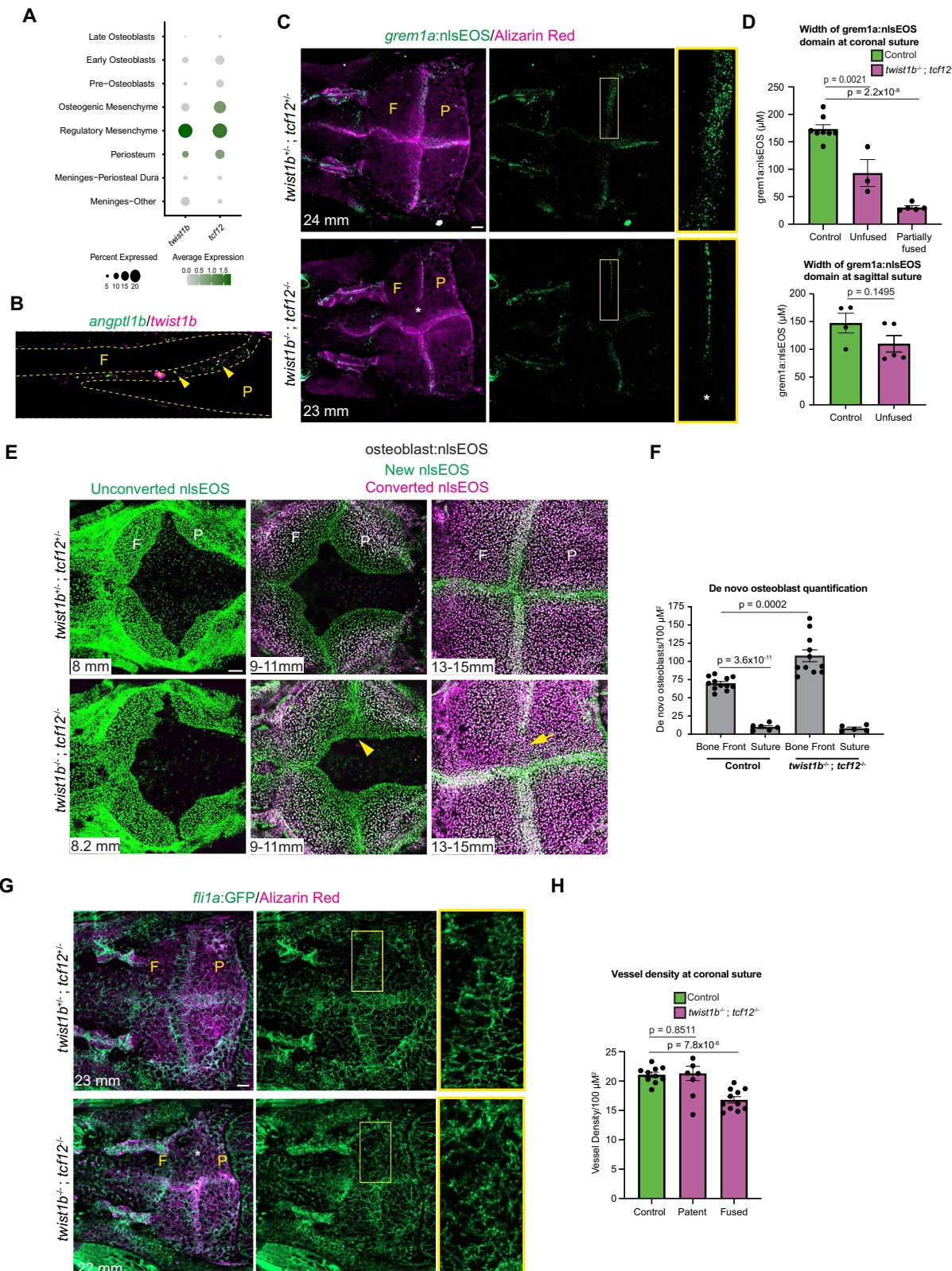

with imaging-based lineage tracing in zebrafish, we demonstrate that bone-forming activity is largely restricted to suture edges. In contrast, lineage tracing of *grem1a* + MSM cells, combined with mutations in BMP antagonists selectively expressed in this population, suggests the presence of a dedicated subpopulation of suture cells that regulate the rate and pattern of bone formation rather than directly contribute to osteoblasts. In addition, we show that the zebrafish homologs of the

Saethre-Chotzen syndrome genes *TWIST1* and *TCF12* are selectively expressed in and required to establish the *grem1a* + MSM population, supporting roles of these transcription factors outside of osteogenic cells in regulating calvarial bone growth and suture maintenance[5,10,13,35].

Consistent with shared genetic requirements for suture formation in zebrafish and mouse[13], we found that most coronal and frontal suture cell types are conserved across species. While osteogenic

**Fig. 7 | Loss of mid-suture mesenchyme and misregulated osteogenesis in *twist1b; tcf12* mutants. A** Dot plot shows enrichment of *twist1b* and *tcf12* expression in the regulatory mesenchyme cluster. **B** Double in situ analysis of adult coronal suture for *twist1b* and *angptl1b*. Dotted lines outline the frontal (F) and parietal (P) bones. In situs were performed in biological triplicate. Arrowheads mark double positive cells within the suture mesenchyme. Asterisk indicates autofluorescence from blood vessel. **C** Confocal images show *grem1a*:nlsEOS suture mesenchyme expression relative to the Alizarin Red+ frontal (F) and parietal (P) bones of control (*twist1b*[+/−];*tcf12*[+/−]) and *twist1b*[−/−];*tcf12*[−/−] mutant fish. Asterisk marks fused region of coronal suture, and boxes denote magnified regions. **D** Quantification of width of *grem1a*:nlsEOS domain in controls and in mutants sides with patent or partially fused coronal sutures. $N = 8$ control coronal sutures, $N = 3$ mutant patent coronal sutures, $N = 5$ partially fused coronal sutures, $N = 4$ control sagittal sutures, $N = 5$ mutant sagittal sutures. **E** Serial confocal imaging of individual control (*twist1b*[+/−]; *tcf12*[+/−]) and *twist1b*[−/−]; *tcf12*[−/−] mutant fish. The same fish were imaged before

conversion of *osteoblast*:nlsEOS at approximately 8 mm, converted at 9 mm and imaged at 11 mm, and then converted again at 13 mm and imaged at 15 mm. Arrowhead marks the aberrant bone front, and arrow marks the fused region of the coronal suture. **F** Quantification of osteoblast addition at bone fronts and coronal sutures. $N = 12$ calvarial bones per genotype imaged from 9–11 mm, $N = 6$ patent coronal sutures per genotype imaged from 13–15 mm. Error bars report standard error of the mean, and statistics were performed using unpaired $t$ tests. **G** Confocal images show *fli1a*:GFP vessels relative to the Alizarin Red+ frontal (F) and parietal (P) bones of control (*twist1b*[+/−];*tcf12*[+/−]) and *twist1b*[−/−];*tcf12*[−/−] mutant fish. Asterisk marks fused region of coronal suture, and boxes denote magnified regions. $N = 5$ controls and $N = 8$ mutants. **H** Quantification of density of *fli1a*:GFP vasculature in controls and mutants at patent or fused coronal sutures. Scalebars: 100 μM. *P*-values were calculated using a two-tailed non-parametric Student's $t$ tests. Error bars represent S.E.M. Source data are provided as a Source Data file.

trajectories were similar, reflecting the deep conservation of bone formation pathways across vertebrates, we observed less complexity of meningeal and ectocranial cell types in fish. For example, we did not identify a correlate of the *Scx*+ ligament-like population above the mouse coronal suture. Although we cannot rule out experimental differences in tissue isolation between species, this might reflect species-specific differences in the biomechanical requirements of cranial sutures, such as the unique compression of the skull during mammalian parturition.

Our work also reveals how cell types present during initial calvarial bone growth may contribute to the mature sutures. Previous reports had suggested cells migrate from the supraorbital ridge to populate the mouse coronal suture[36], yet here we uncovered a mesenchymal signature shared between cells at the leading edge of the bone fronts and those in sutures. Shared genes include *Six2/six2a*, *Pthlh/pthlha*, and *Erg/erg* (and the related zebrafish ETS factor *fli1a*, previously called *Ergb* in mouse). At the mouse coronal suture, our previous lineage tracing had shown that embryonic *Six2*+ mesenchymal cells contribute to both the mid-suture mesenchyme and growing skull bones[5], and *Pthlh* has been shown to be a marker of skeletal stem cells in murine endochondral bones[37]. This suggests that suture mesenchyme may arise, at least in part, from cells at the leading edges of the growing bones. Future lineage tracing experiments will be needed to determine the relative contributions of migratory and leading edge cells to suture mesenchyme.

Consistent with previous studies in mouse[31], our precise nlsEOS-based analysis of new osteoblast addition in living zebrafish shows that bone formation is largely confined to the tips of the overlapping bones at suture edges. In contrast, a subset of mid-suture mesenchyme acquires a distinct gene expression signature, with lineage tracing with *grem1a*:nlsEOS showing minimal contribution to new osteoblasts over one month. While *Grem1* marks skeletal stem cells within the long bones of mice[38], the *grem1a*+ population in the zebrafish mid-suture region does not appear to be a progenitor for new bone. However, RNA velocity analysis shows directional flow from *grem1a*+ MSM to *hyla4*+/*fli1a*+ OM, suggesting a possible latent capacity for MSM to contribute to bone under non-homeostatic conditions such as injury. In contrast to the non-osteogenic properties of *grem1a*+ MSM, pseudotime analysis points to *hyla4*+/*fli1a*+ OM as a likely skeletal stem cell population, although future lineage tracing will be required to confirm this. While OM markers are enriched at the suture edges where the majority of new osteoblasts form, we also detected OM marker expression in a few cells within the mid-suture region. Our findings are therefore consistent with heterogeneity of mid-suture mesenchyme, with *grem1a*+ cells in this domain potentially regulating the behavior of neighboring skeletal stem cells both within the mid-suture region and suture edges.

Several pieces of evidence suggest that the *grem1a*+ MSM subpopulation may have an important regulatory function. Upon their

transition from the bone fronts to the mid-suture region, these cells upregulate a number of genes encoding BMP and Tgfb antagonists, Angiopoeitins, and other factors. BMP signaling has well established roles in stimulating bone formation[39], and previous studies had shown that misexpression of the BMP antagonist *Nog* was able to prevent normal fusion of the mouse posterior frontal suture[40]. Loss of *Bmpr1a* in *Axin2*+ skeletal stem cells impairs self-renewal and leads to ectopic bone formation and craniosynostosis[41], and expression of constitutive active *Bmpr1a* in neural crest lineage cells leads to premature fusion of the metopic suture[42]. These data demonstrate a critical need for tight control of BMP signaling at cranial sutures. Here we show that combinatorial loss of *grem1a*, *nog2*, and *nog3*, three BMP antagonists selectively upregulated and enriched in *grem1a* + MSM, results in altered suture morphology, misregulated calvarial bone growth, and ectopic bone formation. We also find increased proportions of cells with pSmad1/5, a marker of BMP activity, in BMP antagonist mutants, consistent with a local role of MSM-derived BMP antagonists in suture regulation. However, future studies that specifically remove these antagonists within suture mesenchyme will be needed to definitively demonstrate requirements for BMP antagonists in the MSM population for suture regulation[43–46]. It will also be interesting to assess the effect of ablating the *grem1a* + MSM population on suture formation and maintenance, as well as identifying the upstream signals that activate the MSM-specific expression program during suture establishment. Whereas previous reports had not resolved distinct osteogenic and non-osteogenic subsets at mouse sutures[5], we find that expression of inhibitors of BMP and TGF-beta signaling and the Angptl family may be common features of mid-suture mesenchyme in both zebrafish and mouse.

Our findings also provide insights into the etiology of craniosynostosis in Saethre-Chotzen syndrome. Expression of the zebrafish homologs of the genes mutated in Saethre-Chotzen syndrome, *twist1b* and *tcf12*, is enriched in the MSM population. In *twist1b; tcf12* mutants, the *grem1a* + MSM population is reduced, even in sutures that remain patent, consistent with studies in mice showing loss of *Grem1* suture expression in *Twist1*[+/−]; *Tcf12*[+/−] mutants[13]. In both *twist1b; tcf12* mutants[13] and fish with loss of MSM-enriched BMP antagonists (*grem1a*, *nog2*, *nog3*), we observed an initial acceleration of bone growth and ectopic bone formation, followed by a later stalling of bone growth. The failure of frontal and parietal bones to properly overlap at the coronal suture is also a shared phenotype of *grem1a*[−/−]; *nog2*[−/−]; *nog3*[+/−] mutant zebrafish and *Twist1*[+/−]; *Tcf12*[+/−] mutant mice. However, we failed to observe the coronal suture fusions of *twist1b; tcf12* mutants in animals with combinatorial BMP antagonist loss. This could be due to the juvenile lethality of these mutants, our inability to homozygose the *nog3* allele, redundancy with other BMP antagonists expressed in the MSM population (e.g. *fstl3*), and/or contribution of other MSM genes to suture patency. For example, we note MSM enrichment of *tgfbi* and *bambia*, which both negatively regulate the

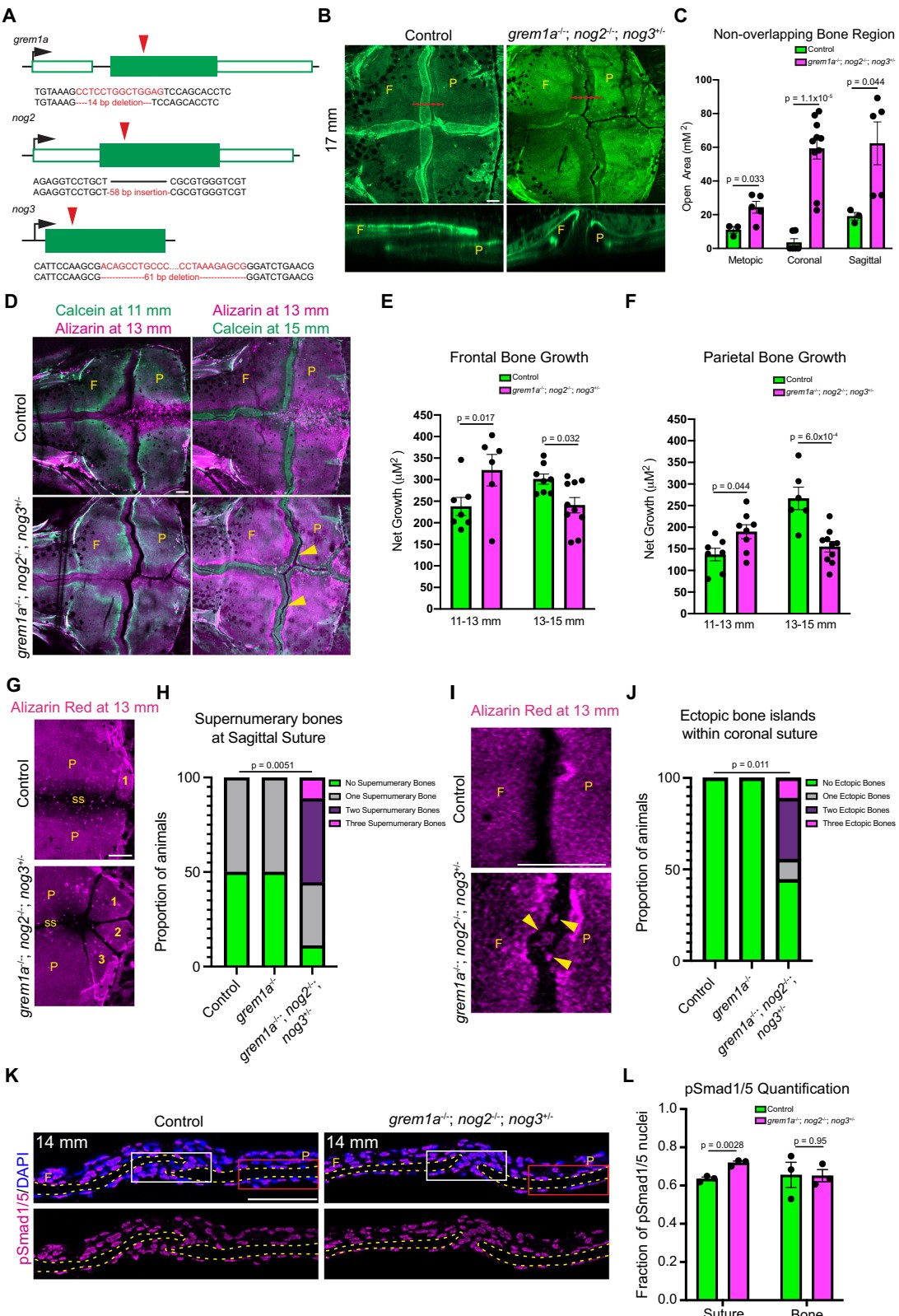

Tgf beta signaling pathway. Loss of an MSM-like population in the mid-suture region could also contribute to the extensive cell mixing across suture boundaries observed in mouse mutants[47,48]. In addition, the MSM population expresses several members of the Angptl family that have well known roles in angiogenesis[49], with loss of the MSM population in *twist1b; tcf12* mutants correlating with reduced density of vasculature networks at sutures. Interestingly, a loss of lymphatic vessels at the coronal sutures of *Twist1*[+/−] mice has recently been linked to neurocognitive impairments[50]. In the future, it will be important to determine the extent to which Twist1 and Tcf12 directly regulate the expression of MSM-specific genes, especially those induced during suture establishment.

**Fig. 8 | Deletion of BMP antagonists alters bone growth and promotes ectopic bone formation. A** Schematic of mutant alleles generated using CRISPR/Cas9. Red arrow indicates site of mutation relative to protein-coding regions (green boxes). **B** Confocal imaging of Calcein-stained calvaria show impaired suture formation in *grem1a⁻ᐟ⁻* ; *nog2⁻ᐟ⁻* ; *nog3⁺ᐟ⁻* zebrafish compared to controls. Dotted lines represent regions of the orthogonal sections at the coronal suture shown below. Frontal (F) and parietal (P) bone. **C** Quantification of non-overlapping regions in the calvaria. *N* = 3 for controls, *N* = 5 for mutants. **D** Representative imaging of individual control and mutant zebrafish after repeated staining and imaging using Calcein green and Alizarin red stains. Arrowheads indicate non-overlapping bones at the coronal suture. **E, F** Quantification of area of bone growth between 11–13 mm and 13–15 mm at frontal and parietal bones. *N* = 7 control frontal bones, *N* = 8 mutant frontal bones at 11–13 mm. *N* = 6 control frontal bones, *N* = 10 mutant frontal bones at 13–15 mm. *N* = 7 control parietal bones, *N* = 10 mutant parietal bones at 11–13 mm. *N* = 6 control

parietal bones, *N* = 10 mutant parietal bones at 13–15 mm. **G** Confocal imaging of supernumerary bones (numbered) at the sagittal suture (ss) between parietal bones in control and mutant zebrafish. **H** Quantification of supernumerary bones at the sagittal suture. *N* = 8 for control and mutant fish. **I** Confocal imaging of control and mutant coronal sutures reveal ectopic bone islands at the mutant coronal suture. Arrowheads mark ectopic bone islands between frontal and parietal bones. **J** Quantification of ectopic bone islands at the coronal suture. *N* = 8 for control and mutant fish. **K** Immunofluorescence for pSmad1/5 at the coronal suture of control and mutant zebrafish. Dotted line outlines bones. White box indicates quantified area for the coronal suture and red box indicates quantified area for parietal bone. **L** Quantification of fraction of pSmad1/5+ nuclei at coronal sutures or along the parietal bone. *N* = 3 for control and mutant fish. Scalebars: 100 μM. *P*-values were calculated using a two-tailed non-parametric Student's *t* tests. Error bars represent S.E.M. Source data are provided as a Source Data file.

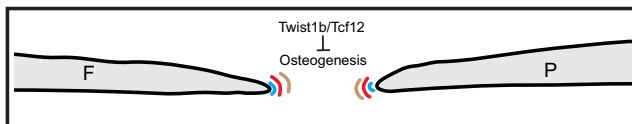

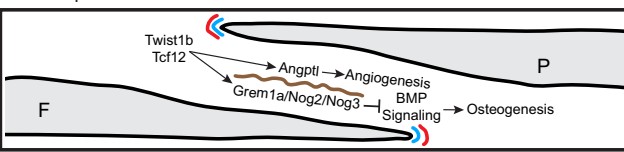

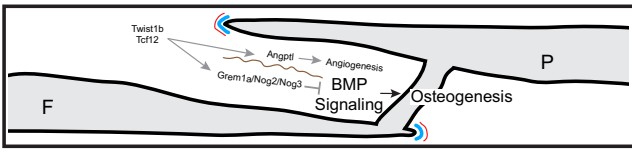

**Fig. 9 | Model for cranial suture formation.** Model for cranial suture formation shows approximating bone fronts with dedicated mesenchyme subsets preceding bone edges, followed by the establishment of overlapping bones and the emergence of the mid-suture mesenchyme program. In a zebrafish model for Saethre-Chotzen syndrome, loss of *twist1b* and *tcf12* fusion leads to a failure to establish the mid-suture mesenchyme program that provides important regulatory factors to restrict osteogenesis and promote angiogenesis at cranial sutures.

## Methods

### Zebrafish
All experiments were approved by the Institutional Animal Care and Use Committee at the University of Southern California (Protocol #20771) and the University of California, Los Angeles (ARC-2022-044). Published lines include Tg(−3.*5ubb*:loxP-EGFP-loxP-mCherry)[51], Tg(Mmu.*Sox10*-Mmu.Fos:Cre)[zf384 52], Tg(*actab2*:loxP-BFP-STOP-loxP-dsRed)[sd27 53], Tg(*fli1a*:eGFP)[y1 54], Tg(*sp7*:mCherry)[zf604 55], Tg(Hsa.*RUNX2*-Mmu.Fos:EGFP)[zf259 55], TgBAC(*mmp9*:EGFP)[tyt206 29], *cdh1*:mlanYFP[xtl7Tg 56], Tg(*sp7*:EGFP)[b1212 57], *tcf12*[el548 13], and *twist1b*[el570 13]. Two transgenic lines were created using CRISPR/Cas9-based genomic integration of nlsEOS: *grem1a*:nlsEOS (guide RNAs targeting the first exon: 5'-AATGGCGCCTTGAAATCCCC-3', 5'-CAGTCACCGCAGACACCGC-3'), *angptl1b*:nlsEOS (guide RNAs targeting the first intron: 5'-GTCGGTGTGTCGGAGACTGT-3', 5'-TTACACGCTCTCTGCACTCC-3'). One-cell embryos were co-injected with guide RNAs (200 ng/μL), mbait guide RNA (200 ng/μL), Cas9 mRNA (150 ng/μL), and a mbait-

nlsEOS plasmid (20 ng/μL)[34,58]. Founders were identified by screening progeny for nlsEOS fluorescence. The *grem1a*:nlsEOS reporter recapitulated endogenous *grem1a* expression (Fig. 5C). In contrast, *angptl1b*:nlsEOS expression differed from endogenous *angptl1b* mRNA by being enriched in osteoblasts but not suture mesenchyme (Supplementary Fig. 6C); hence we describe this line as *osteoblast*:nlsEOS in the text. Knockout alleles were generated using CRISPR/Cas9 as previously described[59] for *grem1a* (sgRNA, 5'-GGTGCTGGACTCCAGC CAGG-3'), *nog2* (sgRNA, 5'-GGAGCACGACCCACGCGAGC-3'), and *nog3* (sgRNA, 5'-CGCTCTTTAGGGTCCAGTAC-3'). Genotyping primers are provided in Supplementary Data 5.

### RNAscope in situ hybridization and immunofluorescence
Juvenile and adult fish were measured and fixed individually in 4% PFA overnight at 4°C. Heads were decalcified for 1–2 weeks and processed by paraffin embedding as previously described[13]. RNAscope reagents were purchased from Advanced Cell Diagnostics, and experiments were performed using the RNAscope Fluorescent Multiplex V2 Assay according to the manufacturer's protocol for formalin-fixed paraffin-embedded (zebrafish tissue) and for fixed frozen (mouse) sections. The following probes were used for this study: Channel 1, Mm-*Angptl2*, Mm-*Nog*, dr-*grem1a*, dr-*ifitm5*, dr-*mmp9*, dr-*prrx1a*, dr-*twist1b*; Channel 2, Mm-*Sp7*; Channel 3, Mm-*Tgfbi*, dr-*angptl1b*, dr-*fli1a*, dr-*foxc1b*, dr-*nog3*, dr-*six2a*, dr-*spp1*, dr-*zic3*, dr-*pah*; Channel 4, *EOS*, dr-*hyal4*, dr-*ogna*, dr-*angptl2b*, dr-*crhbp*. Immunofluorescence was performed using the same antigen retrieval reagents in the RNAscope Fluorescent Multiplex kit, and slides were blocked for 1 h in 10% goat serum, stained with primary overnight (1:100, Cell Signaling #9516) and stained with secondary (1:250, Thermo, A-11011) and DAPI for 1 h.

### Neural crest transplantation
Gastrula-stage embryos (6 h post-fertilization) were collected and donor ectoderm from the animal cap of *ubb*:loxP-EGFP-loxP-mCherry embryos was transplanted into the neural crest precursor domain of wild-type hosts as previously reported[60]. Successful transplantation was confirmed by screening for GFP expression in the face at 5 dpf. Fish were raised to juvenile stages and imaged before frontal bones completely overlap.

### Imaging
All imaging was performed on a Zeiss LSM800 or a Zeiss LSM980 microscope using ZEN software. For repeated live imaging experiments, fish were anesthetized in Tricaine, mounted, and imaged using a 2.5X, 5X or 10X objective. For whole calvaria nlsEOS conversion experiments, live fish were transferred to a 6- or 12-well dish and exposed to a handheld UV light (UV Flashlight Black Light, 3-in-1 Magnetic Flashlight Rechargeable, AdamStar) for 5–30 min. Converted fish were individually housed and re-imaged for up to 2 months following initial conversion.

## Generation and analysis of scRNA-seq datasets

We dissected the calvaria from 20 juvenile (9-11 mm) *cdh1*:mlanYFP (one dataset) or *sp7*:GFP (one dataset) fish, and separately 10 adult (22–25 mm) *cdh1*:mlanYFP (one dataset) or Tubingen (one dataset) fish, away from brain tissue in Ringer's solution. Calvaria were mechanically minced with a razor and all tissue was transferred to a 1.5 mL tube for enzymatic dissociation (0.25% trypsin (Life Technologies, 15090-046), 1 mM EDTA, and 400 mg/mL Collagenase D (Sigma, 11088882001) in PBS). Tissue was incubated on a nutator at 28.5 °C for ~45 min and the enzymatic reaction was stopped by adding 6X stop solution (6 mM $CaCl_2$ and 30% fetal bovine serum (FBS) in PBS). Cells were pelleted by centrifugation (300 $g$) at 4 °C, rinsed in suspension media (1% FBS, 0.8 mM $CaCl_2$ (Sigma-Aldrich, St. Louis, MO) in phenol red-free Leibovitz's L15 medium (Life Technologies)), and resuspended in a final volume of 500 uL. DAPI was added to resuspended cells, and cells were fluorescence-activated cell sorted to isolate live cells that were DAPI negative and, when applicable, *cdh1*:mlanYFP/*sp7*:GFP negative to deplete epithelial cells.

scRNAseq libraries were prepared using 10X Genomics Chromium Single Cell 3' Library and Gel Bead Kit v.2 according to the manufacturer's instructions and sequenced using Illumina NextSeq or HiSeq machines at a depth of at least 50,000 reads per cell for each library. Juvenile and adult stages were performed in replicates, and sequenced data was aligned using Cellranger v3.0.0 from 10X Genomics against the GRCz11 genome. All parameters were set to their default values. Data analysis was performed with Seurat version 4.1.1[19], and datasets were filtered by nFeature_RNA > 200, nFeature_RNA < 2500, and percent.mt <25. Filtered cells were normalized with SCTransform and datasets were integrated based on the tutorial "Integration and Label Transfer" from Seurat (https://satijalab.org/seurat/archive/v3.0/integration.html). The data were processed using the FindNeighbors() and FindClusters() functions and data were visualized with UMAP (30 principal components). The original dataset was first resolved at a resolution of 0.2 to identify the overall cell types contained within the dataset. The connective and skeletogenic subset was visualized at a resolution of 1.0 and the prrx1a+ mesenchyme and osteoblast subset was visualized at a resolution of 0.8, as these resolution values demonstrated the highest number of clusters with transcriptionally unique signatures. The osteogenic subset of the integrated Seurat object was converted into a Monocle cell dataset and Monocle was performed following the Monocle3 recommended parameters (https://cole-trapnell-lab.github.io/monocle3/docs/trajectories/#learn-graph). The prrx1a+ mesenchyme and osteoblast subset was analyzed using the veloctyo[61] and CellChat packages[27]. Scores for regulatory mesenchyme in mouse datasets were assessed and analyzed as previously described[5]. Mouse datasets were downloaded from Facebase (www.facebase.org, coronal suture, Accession: FB00001236; frontal suture, Accession: FB00001013).

## Live bone staining

Fish were stained with Calcein Blue (3 mg/30 ml, Molecular Probes C481) or Alizarin Red (1 mg/30 ml, Sigma A5533) overnight in the dark and washed in fish system water for 1–2 h each before imaging. For repeated live imaging experiments, fish were anesthetized and measured to stain for net growth every 2 mm between 11-15 mm as previously described[13]. Briefly, 11 mm fish were stained overnight in Calcein Green stain (1 mg/10 mL, Molecular Probes C481) and washed for at least one hour before being returned to individual housing. At 13 mm, fish were stained with Alizarin Red S (1 mg/30 ml), washed for at least one hour and imaged on a Zeiss LSM980 to detect Calcein and Alizarin Red S signal. Fish were again returned to individual housing and grown to 15 mm and stained with fresh Calcein. After a one-hour wash, fish were again imaged to detect Calcein and Alizarin Red S signal.

## Adult skeletal preparations

Skeletal preparations were performed as previously described[13], and skullcaps were dissected from the skull and imaged on a Zeiss Stemi with Zeiss Labscope software in 100% glycerol.

## Quantification

For *osteoblast*:nlsEOS experiments, regions for quantification were defined by manually drawing a ROI across bone fronts or cranial sutures using ImageJ. The channels were then split and nuclei within each channel were counted manually. De novo osteoblasts were defined as green only cells. Masked images were generated using Illustrator by converting double positive and magenta osteoblasts to a white background to mask previously existing osteoblasts. For *grem1a*:nlsEOS experiments, the width of the nlsEOS+ zone was measured across cranial sutures (at each end and in the middle) using ImageJ and averaged. For quantification, *sp7*:mCherry; *grem1a*:nlsEOS double-positive cells were manually counted from 40X images through the Z-stack using CellCounter in ImageJ. For live bone staining quantification, the area of Calcein and Alizarin Red S stained bones were measured using the freehand selection on ImageJ, and the outer bone areas were subtracted from the inner bone area to determine net area growth. Vessel density was quantified using the Vessel Analysis plugin[62] after defining a 250 $\mu M^2$ region at patent and fused coronal sutures. For phosphorylated Smad1/5 quantification, an approximately 120 μM by 40 μM rectangle was drawn around the coronal suture to include bone fronts and suture mesenchyme or around the parietal bone, excluding the top two layers of DAPI+ cells to avoid quantifying skin. DAPI positive and pSmad1/5 positive nuclei were manually counted using CellCounter in ImageJ. Statistical analyses were performed using Prism.

## Reporting summary

Further information on research design is available in the Nature Portfolio Reporting Summary linked to this article.

## Data availability

The scRNAseq datasets and the processed Seurat rds objects in this study have been deposited in the Gene Expression Omnibus (GEO) database under accession code GSE223147. Source data are provided with this paper.

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

## Acknowledgements

The authors thank Megan Matsutani and Jennifer DeKoeyer Crump for fish care. We thank Stephen Twigg for input on the initial manuscript. We thank the Children's Hospital Los Angeles Molecular Pathology Genomics Core for next-generation sequencing and the USC Stem Cell Flow Cytometry Facility for cell sorting.

## Author contributions

D.T.F., C.T., R.M., and J.G.C. conceived and designed the study. D.T.F., H.C., C.A., and J.H.T. carried out the single-cell and bioinformatic analysis. D.T.F and J.D. performed repeated live imaging experiments. D.T.F. and P.X. generated mutant lines. J.G.C. and D.T.F. supervised the research. D.T.F. and J.G.C. wrote the paper. NIH (5R01DE026339, J.G.C.), NIH F31 Fellowship (C.A.), HHMI Hanna H. Gray Fellows Program (D.T.F), Burroughs Wellcome PDEP (D.T.F.), Society of Developmental Biology Choose Development! Fellow (J.H.T).

## Competing interests

The authors declare no competing interests.
