## [Peer Review file · Nature Communications]

REVIEWER COMMENTS

Reviewer #1 (Remarks to the Author):

General comments:

This manuscript entitled “Cranial sutures spatially restrict osteogenesis to sustain skull expansion” examined cranial sutures mainly using single-cell transcriptomics and imaging-based cell tracing in zebrafish. The results indicated new bone formation restricted to suture edges supporting many prior reports. However, it has been well established that mid-suture consists of stem/progenitor cells, and osteogenic activities are known to be at the front of calvarial bones. This part of the work is confirmatory in nature and does not add anything new to our current knowledge. The authors then described a non-osteogenic *grem1a*⁺ cell population and proposed they act as regulatory mesenchyme cells. The data showing *grem1a*⁺ cells did not contribute to bone formation are questionable as they contradict previous findings showing *Grem1*⁺ are skeletal stem cells with bone, cartilage, and reticular stromal potential. Using the zebrafish model of coronal synostosis of Saethre-Schotzen Syndrome (*twist1b*; *tcf12* mutant), the authors found alteration of *grem1a*⁺ cells, thus implying they are regulatory mesenchyme cells modulating the osteogenic activity. Unfortunately, the results presented are correlative observations and did not in any way indicate that *grem1a*⁺ cells have the regulatory function to the skeletal stem and progenitor cells. Due to the lack of direct evidence, the conclusion lacking rigor is highly premature. Overall, this is mainly a confirmatory study lacking novelty and critical functional examination of the identified *grem1a*⁺ cells.

Specific comments:

In the Abstract, several reports have demonstrated that cranial sutures contain SSCs to drive bone growth so the use of “proposed” is inaccurate (lines 27).

In lines 28-29, there is clear evidence showing the presence of skeletal progenitors is required for suture patency so the use of “unresolved question” is incorrect.

In lines 44-46, the authors appear to overly state their findings based on correlative analyses. No data supports mid-suture regulatory mesenchyme restricts the osteogenic activity of skeletal progenitors to the suture edges as the experimental design does not directly examine the regulatory function of *grem1a*⁺ cells. The evidence showing the regulation of skeletal progenitors by *grem1a*⁺ cells is missing in this study.

In lines 60-61, several reports have shown the precise location of the osteogenic stem cells relative to the suture, so the statement of their undefined precise location is wrong.

In the Introduction, the authors need to acknowledge prior work performed by others, e.g., no references for their two sentences about neural crest and mesoderm origin, CTSK+ stem cells, and BMP antagonist and signaling in suture patency.

In Fig 1, several populations of immune cells occupy more than 40% of the total cells but there are no descriptions of these cells. Cell type annotation is incomplete. Also, are immune cells known to be major a cell type found in cranial sutures? Or, do these cell populations come from contamination during cell sorting?

In addition, what resolution was used for clustering analysis?

The statement in lines 164-165 is inaccurate as a few cell tracing analyses have demonstrated the origin of bone-forming cells comes from mid-suture.

Runx2+ and Sp7+ osteogenic cells near the bone fronts have been well established in the literature so there is nothing novel about the observation shown in Fig 2. Furthermore, cell tracing analyses have demonstrated that SSCs in mid-suture gradually expand into the lateral area and bone fronts eventually occupy the entire suture mesenchyme. These results are confirmatory of previous findings.

In Fig 2, the analyses seem incomplete as there is no coronal suture data in juveniles and no frontal bone data in adults.

In line 188, Mmp9 is an angiogenic factor important for osteogenesis during endochondral ossification. But is there any evidence for Mmp9 as a pre-osteoblast marker? No reference was cited for mmp9 expression specific to pre-osteoblast cells. The information stated on line 293 from the corresponding references cannot be found.

Along the same line, cranial bone formation involves intramembranous ossification but not endochondral ossification. Why does Mmp9 express here?

In Fig 3, the authors used pseudotime analysis with Monocle3 to identify the trajectory from osteogenic mesenchyme to late osteoblasts (line 190). But it is not clear which root the authors select. How was the root determined?

Have the authors performed RNA velocity analysis for these clusters?

Although nlsEOS is a new tool, the results obtained are confirmatory of prior publications thus lacking novelty.

Results presented in Figs 1-4 are confirmatory of numerous studies indicating mid-suture consists of stem and progenitor cells and osteogenesis occurs at the bone edge.

The analysis of grem1a+ cells suggests that they are non-osteogenic, and do not contribute to bone formation. However, a report by Worthley et al has demonstrated that Grem1+ cells are SSCs contributing to bone formation in mice. There is no mention of this Cell paper published in 2015.

The authors need to discuss the discrepancy between their results and previous findings regarding the Grem1+ cell population.

In Fig. 6C, did the authors perform photoconversion only on the boxed region? It was not clear how this experiment was performed.

In Fig 7, how do the authors know the grem1a+ cells are regulatory mesenchyme? There is no examination of their effects on osteogenic cell types.

The authors should consider the co-localization study of twist1b and tcf12 genes with RM markers, e.g., grem1a, nog2, nog3.

The reduction of grem1a+ cells in the fish craniosynostosis model twist1b; tcf12 is a correlative observation. The results did not in any way indicate that grem1a+ cells have any regulatory function.

The direct testing of *grem1a*⁺ cells on the regulation of osteogenic cells is missing but absolutely required for the conclusion. The authors need to perform functional studies of these cells. Due to the lack of direct evidence, the results have been overly interpreted, and the conclusion is premature at this stage.

Reviewer #2 (Remarks to the Author):

The manuscript by Farmer et al. analyzes scRNAseq data from two different stages in zebrafish skull development, with a focus on defining cell populations at the forming sutures between bones of the calvarium. They distinguish two mesenchyme populations in the sutures: osteogenic, in close association with the edges of the bones; and regulatory, in the mid-sutures. Pseudotime analysis suggests a progression of cells from osteogenic mesenchyme to late osteoblasts, an interpretation supported by in situ localization of gene expression and by direct lineage tracing in both juvenile and adult fish.

Their findings contrast with prior mouse data suggesting that mesenchymal cells of the mid-suture contain osteoblast precursors, or that the underlying dura can give rise to new osteoblasts. However, they analyze previously acquired scRNAseq data sets from mouse sutures to show that many of the same cell types and clusters are present in the mouse, supporting conservation of suture organization and function between the two species. They localize expression in situ for several fish orthologues of mouse markers, also supporting broad conservation of cell types. Finally, they propose a model where the mid-suture mesenchyme regulates the rate of bone growth and generation of new osteoblasts. Consistent with their model, they show that markers of the regulatory mesenchyme population are reduced or absent in a zebrafish model of Saethre-Chotzen syndrome, correlating with loss of the sutural gap and fusion of the bones.

Overall, the data are of excellent quality and support their conclusions. In particular, they have several lines of evidence supporting their main conclusion, that the osteoblast precursors reside near the edges of the growing skull bones. Without direct lineage tracing, they cannot conclude that the same is true in the mouse, but their data certainly suggest that it is. They also propose a model for the pathogenesis of Saethre-Chotzen syndrome, where the loss of regulatory mesenchyme is the underlying cause, rather than a cell-autonomous defect in the osteoblasts. They offer compelling data to support the model, although cell-autonomous effects could be key in other genetic craniosynostosis cases.

One minor criticism is that a few figure panels, currently showing single-color fluorescence, would be easier to see in greyscale (Fig 2D, E; Fig 3F; Fig 5C; Fig 6A; Fig 7B). Also, in the graphical abstract, the line colors of the osteogenic and regulatory mesenchyme are difficult to distinguish.

Reviewer #3 (Remarks to the Author):

In this study, the authors aimed to characterize suture cells in zebrafish using calvaria at juvenile bone front stages and adult stages by utilizing single-cell transcriptomics and integrating their findings with similar studies in mice. Through this approach, they identified conserved cell populations and validated them through in vivo expression analysis with several genetic tools. Furthermore, they discovered a novel non-osteogenic population of cells located in the mid-suture region after bones overlap, which they named regulatory mesenchyme (RM). While genetic analysis suggested that this population did not have osteogenic potential, the authors propose that it plays a role in restricting osteogenesis to sustain skull expansion. Although the study is comprehensive and the data are convincing, there are some issues with the identification and characterization of RM population that needs to be addressed.

Main comment 1

The authors named this non-osteogenic mesenchymal cell population regulatory mesenchyme (RM); however, there is currently no direct evidence demonstrating that this population indeed regulates osteogenesis. Although the authors showed that RM cells express genes related to BMP or TGF β signaling antagonists and provided insights into the relevance of this cell type to Saethre-Chotzen Syndrome through their experiments with *twist1b*; *tcf12* mutant zebrafish, a targeted cell ablation assay is necessary to fully support their claim.

Main comment 2

Related to the comment 1, performing a ligand-receptor analysis can provide additional insights into potential signaling interactions between the regulatory mesenchyme (RM) and osteoblast lineages. The authors need to address this assay. Additionally, the authors may consider validating the contributions of these signaling pathways in a bioassay. This would provide further evidence for the role of RM in regulating osteogenesis.

Main comment 3

The study focuses on comparative analysis between zebrafish and mouse, but it is not clear whether the distinct gene expression signature of the RM population is conserved in both species. While the authors thoroughly characterized the RM population in zebrafish, they need to provide

more information on the RM population in mouse, including its characterization in single-cell analysis and its spatial distribution in mouse calvaria suture. Addressing these points will be important for establishing the conservation of the RM population between species.

Main comment 4

Further computational analysis in the single cell RNA-seq data potentially sheds light on how the RM population emerges during skull development. By performing trajectory analysis and gene regulatory network analysis, the authors can identify the intermediate cell states and gene regulatory networks that give rise to the RM population. This will provide important insights into the developmental origin of the RM population and its role in regulating osteogenesis. The authors should consider performing such analyses and discussing its potential implications in their study.

Minor comment

I would recommend to omitting Fig. 3H, because current study does not fully support this scheme and more importantly, illustration of RM, a main finding in this study, is missing in the figure.

REVIEWER COMMENTS

Reviewer #1 (Remarks to the Author):

General comments:

This manuscript entitled “Cranial sutures spatially restrict osteogenesis to sustain skull expansion” examined cranial sutures mainly using single-cell transcriptomics and imaging-based cell tracing in zebrafish. The results indicated new bone formation restricted to suture edges supporting many prior reports. However, it has been well established that mid-suture consists of stem/progenitor cells, and osteogenic activities are known to be at the front of calvarial bones. This part of the work is confirmatory in nature and does not add anything new to our current knowledge. The authors then described a non-osteogenic *grem1a*⁺ cell population and proposed they act as regulatory mesenchyme cells. The data showing *grem1a*⁺ cells did not contribute to bone formation are questionable as they contradict previous findings showing *Grem1*⁺ are skeletal stem cells with bone, cartilage, and reticular stromal potential. Using the zebrafish model of coronal synostosis of Saethre-Chotzen Syndrome (*twist1b;tcf12* mutant), the authors found alteration of *grem1a*⁺ cells, thus implying they are regulatory mesenchyme cells modulating the osteogenic activity. Unfortunately, the results presented are correlative observations and did not in any way indicate that *grem1a*⁺ cells have the regulatory function to the skeletal stem and progenitor cells. Due to the lack of direct evidence, the conclusion lacking rigor is highly premature. Overall, this is mainly a confirmatory study lacking novelty and critical functional examination of the identified *grem1a*⁺ cells.

We thank the reviewer for their thoughtful comments. To examine the function of *grem1a*⁺ mid-suture cells, we generated mutants for three *Bmp* antagonists upregulated and specifically expressed in the *grem1a*⁺ mid-suture population - *grem1a*, *nog2*, and *nog3*. While individual mutants have normal sutures, we now reveal in two new figures (Fig. 8 and Fig. S10) that triple mutants display ectopic bones at cranial sutures and an aberrant suture morphology. This strengthens our model that the *grem1a*⁺ mid-suture population has a regulatory function during suture establishment and calvarial bone growth. We have also extensively modified the text to clarify that the mid-suture region is likely heterogeneous and that we cannot rule out a long-term stem cell function of *grem1a*⁺ mid-suture cells due to our short-term lineage tracing technique in zebrafish.

Specific comments:

In the Abstract, several reports have demonstrated that cranial sutures contain SSCs to drive bone growth so the use of “proposed” is inaccurate (lines 27).

We apologize for the unintentional tone of the sentence and have corrected the sentence appropriately.

Lines 28-29:

“Cranial sutures separate neighboring skull bones and contain skeletal stem cells that drive bone growth.”

In lines 28-29, there is clear evidence showing the presence of skeletal progenitors is required for suture patency so the use of “unresolved question” is incorrect.

We agree that this sentence fails to recognize the contributions of several labs demonstrating the necessity of SSCs using various strategies, including the ablation of SSCs. We have corrected the sentence to precisely communicate the objective of our study.

Lines 29-30:

“A key question is how osteogenic activity is controlled to promote bone growth while preventing aberrant bone fusions during skull expansion.”

In lines 44-46, the authors appear to overly state their findings based on correlative analyses. No data supports mid-suture regulatory mesenchyme restricts the osteogenic activity of skeletal progenitors to the suture edges as the experimental design does not directly examine the regulatory function of *grem1a*⁺ cells. The evidence showing the regulation of skeletal progenitors by *grem1a*⁺ cells is missing in this study.

Our new findings that zebrafish lacking all three Bmp antagonists selectively expressed in *grem1a*⁺ mid-suture cells have highly abnormal sutures, ectopic bone, and misregulated osteogenesis now directly support a regulatory function of the *grem1a*⁺ population in suture and calvarial development.

In lines 60-61, several reports have shown the precise location of the osteogenic stem cells relative to the suture, so the statement of their undefined precise location is wrong.

While we agree that there are many excellent studies examining cranial suture residing SSCs, it is often unclear whether only a subset of cells labeled by CreER and other techniques have osteogenic stem cell function. For example, *Gli1*-CreER and *Prrx1*-CreER label quite broad domains in and around mouse sutures. We argue that this is the power of the zebrafish system, where high-resolution imaging can allow us to more precisely define where osteogenic stem cells reside.

Lines 66-67:

“However, it is unclear whether only a subset of cells labeled by these lines have osteogenic stem cell function, and if so where in relation to the suture these reside.”

Discussion, Lines 412-420:

“Our findings in zebrafish may help explain differences in contributions of proposed skeletal stem cell populations in mouse sutures⁶⁻¹⁰. Lineage tracing based on *Gli1* and *Prrx1*, which are expressed broadly in not only mid-suture mesenchyme but also bone tips at suture edges, shows abundant short-term contributions to bone, as well as long-term labeling of bone during homeostasis and repair^{6,8}. In contrast, *Axin2*:CreER more specifically labels mid-suture mesenchyme which shows contributions to substantial bone only after several months⁷. It is therefore possible that the *grem1a*⁺ mid-suture cells we identify contain cells that are the correlates of the *Gli1*⁺/*Axin2*⁺ mid-suture cells in mice, acting as a long-term reservoir of cells that transition over time into a short-term osteoprogenitors.”

In the Introduction, the authors need to acknowledge prior work performed by others, e.g., no

references for their two sentences about neural crest and mesoderm origin, CTSK+ stem cells, and BMP antagonist and signaling in suture patency.

We thank the reviewer for highlighting this oversight. We have included the relevant citations for studies that demonstrate both neural crest and mesodermal contributions to calvarial bones.

Lines 53-55:

“The skull bones arise from mesenchymal cells of neural crest or mesoderm origin that condense and grow apically and laterally to cover the brain^{3,4}.”

We have also expanded our introduction to include the studies employing *Ctsk:Cre* to examine cranial suture SSCs:

Lines 56-66:

“Lineage tracing studies in mouse, for example based on genetic recombination mediated by *Gli1:CreER*⁶, *Axin2:CreER*⁷, and *Prrx1:CreER*⁸, suggest that postnatal sutures house osteogenic stem cells that grow the calvaria throughout adulthood. However, it is unclear whether only a subset of cells labeled by these lines have osteogenic stem cell function, and if so where in relation to the suture these reside. Osteogenic cells at cranial sutures can also be labeled by *Ctsk:Cre*, which labels dedicated stem cells for intramembranous ossification in the periosteum of long bones, suggesting commonalities of bone-forming cells in the calvarium and long bones⁹. Signaling between *Ctsk*-derived stem cells and a newly identified cartilage-promoting *Ddr2+* stem cell at cranial sutures further demonstrates the complex crosstalk necessary to maintain a patent suture¹⁰.”

Additionally, we highlight data supporting BMP antagonist function at cranial sutures in the Discussion:

Lines 430-434:

“Upon their transition from the bone fronts to the mid-suture region, these cells upregulate a number of genes encoding *Bmp* and *Tgfb* antagonists, Angiopoietins, and other factors. *Bmp* signaling has well established roles in stimulating bone formation³⁷, and previous studies had shown that misexpression of the *Bmp* antagonist *Nog* was able to prevent normal fusion of the mouse posterior frontal suture³⁸”

In Fig 1, several populations of immune cells occupy more than 40% of the total cells but there are no descriptions of these cells. Cell type annotation is incomplete. Also, are immune cells known to be major a cell type found in cranial sutures? Or, do these cell populations come from contamination during cell sorting?

While immune cells were not a primary focus of our study, we have now used a recent study describing zebrafish immune cells using scRNA-seq to more comprehensively identify the cell types, including immune cells, captured in our study.

Lines 112-114:

“The immune population, which we do not further investigate in this study, includes cells expressing markers for T-cells, macrophages, and neutrophils, and a cluster with a mixed signature for natural killer cells and T-cells¹⁸ (Fig. S1A).”

In addition, what resolution was used for clustering analysis?

We now provide the resolution used for each dataset in our manuscript within the Methods section.

Lines 537-541:

“The original dataset was first resolved at a resolution of 0.2 to identify the overall cell types contained within the dataset. The connective and skeletogenic subset was visualized at a resolution of 1.0 and the *prx1a*+ mesenchyme and osteoblast subset was visualized at a resolution of 0.8, as these resolution values demonstrated the highest number of clusters with transcriptionally unique signatures.”

The statement in lines 164-165 is inaccurate as a few cell tracing analyses have demonstrated the origin of bone-forming cells comes from mid-suture. *Runx2*+ and *Sp7*+ osteogenic cells near the bone fronts have been well established in the literature so there is nothing novel about the observation shown in Fig 2. Furthermore, cell tracing analyses have demonstrated that SSCs in mid-suture gradually expand into the lateral area and bone fronts eventually occupy the entire suture mesenchyme. These results are confirmatory of previous findings.

We thank the reviewer for raising the point. The intent of the section was to examine the origins of mesenchyme that grow bones prior to suture formation during embryonic development. Contrary to postnatal stages, there are data that suggest that the mesenchyme separating approximating bones does not contribute to bone, while other studies suggest migratory cells move in to contribute to bone fronts. We sought to examine if and how a dedicated mesenchymal population associates with growing bones by examining the presence of cell populations negative for *Runx2* and *Sp7* ahead of the pre-osteoblast bone front population. To our knowledge, this is the first study to demonstrate the existence of a distinct population of *Runx2*-/*Sp7*- mesenchyme at the leading edge of the bone fronts that shares a transcriptional signature with mesenchyme that resides within cranial sutures. We have modified the text to clarify this:

Lines 198-200:

“As the developmental origin and composition of the mesenchyme that supports calvarial expansion prior to suture formation is poorly understood, we first sought to discern the spatial architecture of mesenchyme associated with the growing skull bones.”

We also direct the reviewer to the two sections in our discussion below:

Lines 394-427:

“Our work also reveals how cell types present during initial calvarial bone growth may contribute to the mature sutures. Previous reports had suggested cells migrate from the supraorbital ridge to populate the mouse coronal suture³⁴, yet here we uncovered a mesenchymal signature shared between cells at the leading edge of the bone fronts and those in sutures. Shared genes include *Six2/six2a*, *Pthlh/pthlha*, and *Erg/erg* (and the related zebrafish ETS factor *fli1a*, previously called

Ergb in mouse). At the mouse coronal suture, our previous lineage tracing had shown that embryonic *Six2*⁺ mesenchymal cells contribute to both the mid-suture mesenchyme and growing skull bones⁵, and *Pthlh* has been shown to be a marker of skeletal stem cells in murine endochondral bones³⁵. This suggests that suture mesenchyme may arise, at least in part, from cells at the leading edges of the growing bones. Future lineage tracing experiments will be needed to determine the relative contributions of migratory and leading edge cells to suture mesenchyme.”

“Consistent with previous studies in mouse²⁸, our precise nlEOS-based analysis of new osteoblast addition in living zebrafish shows that bone formation is largely confined to the edges of the sutures, at the tips of the overlapping bones. In contrast, a subset of mid-suture mesenchyme acquires a distinct gene expression signature, with short-term lineage tracing with *grem1a*:nlEOS showing little contribution to new osteoblasts. While *Grem1* marks skeletal stem cells within the long bones of mice³⁶, the *grem1a*⁺ population in the zebrafish mid-suture region does not appear to be a short-term progenitor for new bone. Our findings in zebrafish may help explain differences in contributions of proposed skeletal stem cell populations in mouse sutures⁶⁻¹⁰. Lineage tracing based on *Gli1* and *Prrx1*, which are expressed broadly in not only mid-suture mesenchyme but also bone tips at suture edges, shows abundant short-term contributions to bone, as well as long-term labeling of bone during homeostasis and repair^{6,8}. In contrast, *Axin2*:CreER more specifically labels mid-suture mesenchyme which shows contributions to substantial bone only after several months⁷. It is therefore possible that the *grem1a*⁺ mid-suture cells we identify contain cells that are the correlates of the *Gli1*⁺/*Axin2*⁺ mid-suture cells in mice, acting as a long-term reservoir of cells that transition over time into a short-term osteoprogenitors. In support of this, RNA velocity in zebrafish shows directional flow from *grem1a*⁺ RM to *hyla4*⁺/*fli1a*⁺ OM. Alternatively, mid-suture mesenchyme may be heterogeneous, with *grem1a*⁺ cells having a largely regulatory function and a distinct subpopulation functioning as a long-term stem cell pool for osteogenesis. In support of this, we observe expression of the OM markers *fli1a* and *six2a* in a subset of mid-suture cells that are non-overlapping, at least in part, with RM markers such as *grem1a*. Future tools that allow longer term labeling of mid-suture subpopulations will be required to test this directly.”

In Fig 2, the analyses seem incomplete as there is no coronal suture data in juveniles and no frontal bone data in adults.

We apologize for the confusion. At juvenile stages, the frontal and parietal bones are well separated and there is not yet a coronal suture to examine. We focused on the frontal bone region in juveniles for our analysis. At adult stages, we do show frontal and parietal bones that overlap at the coronal suture.

In line 188, *Mmp9* is an angiogenic factor important for osteogenesis during endochondral ossification. But is there any evidence for *Mmp9* as a pre-osteoblast marker? No reference was cited for *mmp9* expression specific to pre-osteoblast cells. The information stated on line 293 from the corresponding references cannot be found.

We have now added references showing that *mmp9* is a pre-osteoblast marker in the regenerating fin and scale in zebrafish. In single-cell analysis of the mouse coronal suture, we had shown that *Mmp13* is a marker of pre-osteoblasts, primarily in the periosteum. We also note that *Mmp9* and *Mmp13* have been shown to have redundant functions during endochondral ossification in mice, and we also observed co-expression of *mmp13b* with *mmp9* in the osteoprogenitor population in zebrafish. We clarify this background information with modified text and additional references, as

well as highlighting the ability of *mmp9* RNA and *mmp9*:GFP to mark cells at the bone fronts where we show osteoblast formation occurs:

Lines 222-230:

“Using known marker genes as a guide, we identified clusters corresponding to presumptive pre-osteoblasts (*mmp9*⁺, *mmp13b*⁺, *spp1*⁺, *runx2b*⁺), early osteoblasts (*spp1*⁺, *ifitm5*⁺, *sp7*⁺, *bglapl*⁻), and late osteoblasts (*spp1*⁺, *ifitm5*⁺, *sp7*⁺, *bglapl*⁺) (**Fig. 1E; Fig. S6**). Although *Mmp9* has not been described as a marker for pre-osteoblasts during mouse intramembranous ossification, *mmp9* has been shown to mark pre-osteoblasts during regeneration of intramembranous bone in the zebrafish fin²⁶. In addition, *Mmp9* has been shown to have redundant functions with *Mmp13* in mouse long bone development²⁷, we have shown that *Mmp13* marks pre-osteoblasts in the mouse coronal suture⁵, and here we show that *mmp9* and *mmp13b* are co-expressed in pre-osteoblasts of the zebrafish calvarium.”

Lines 239-243:

“Prior to suture formation *mmp9* expression was highly localized to the growing tip of the frontal bone, and after suture formation remained at the tips of the frontal and parietal bones and was generally excluded from the mid-suture region (**Fig. 3E,F**). We confirmed localized expression of *mmp9* to suture edges using an *mmp9*:GFP transgenic line that has been shown to mark pre-osteoblasts in the regenerating fin²⁶ (**Fig. 3G**).”

Along the same line, cranial bone formation involves intramembranous ossification but not endochondral ossification. Why does *Mmp9* express here?

It is an interesting question what role *Mmp9/13* might have in remodeling the extracellular matrix during intramembranous ossification but to address this is beyond the scope of our current study. As noted above, *Mmp13* is expressed in pre-osteoblasts associated with the mouse coronal suture, *mmp9* is expressed in pre-osteoblasts of regenerating fin bone of zebrafish (which regenerates through intramembranous ossification), and *Mmp9* and *Mmp13* have been shown to have redundant functions in other context such as endochondral ossification. We therefore speculate that highly related *Mmp9* and *Mmp13* perform similar functions in zebrafish intramembranous ossification whereas mammals have evolved such that only *Mmp13* is important for intramembranous ossification.

Have the authors performed RNA velocity analysis for these clusters?

We thank the reviewer for this suggestion as new RNA velocity analysis (Fig. 1F) has been quite informative. In particular, strong connections from osteogenic mesenchyme to pre-osteoblasts supports our evidence for this being the primary osteogenic population. In contrast, *grem1a*⁺ regulatory mesenchyme does not connect to pre-osteoblasts, although we note some flow from regulatory mesenchyme to osteogenic mesenchyme. Based on this and prior data in mouse, we have now revised the text to more clearly state that the population we term regulatory mesenchyme may have functions both in restraining bone formation short-term and by providing a more long-term skeletal stem cell reservoir (see above responses for more detail).

Lines 134-141:

“We also performed RNA velocity to infer the directionality of cell state changes based on nascent gene expression. Osteogenic mesenchyme flows toward the committed osteoblast subtypes (pre-osteoblast, early and late osteoblasts), and regulatory mesenchyme has weak connections toward the periosteum and osteogenic mesenchyme (**Fig. 1F**). This analysis supports osteogenic and not regulatory mesenchyme being the main precursor to new osteoblasts in the calvarium, although the weak flow of regulatory to osteogenic mesenchyme might suggest that the regulatory mesenchyme also contains some long-term stem cells for osteogenesis.”

Although nlsEOS is a new tool, the results obtained are confirmatory of prior publications thus lacking novelty.

While we agree that CreER studies in mouse have defined populations of skeletal stem cells in the suture, the osteoblasts and grem1a nlsEOS tools we have developed have allowed us to make a number of new insights. 1) That osteoblast formation is highly localized to the suture edges as opposed to the mid-suture region and that this is conserved in zebrafish. 2) That mid-suture grem1a+ cells are not a major source of new osteoblasts in the short term. 3) That nlsEOS imaging allows very precise quantification of new osteoblast addition across time, allowing us to uncover even subtle osteoblast addition defects in mutants (see Fig. 7 and new Fig. S10).

Results presented in Figs 1-4 are confirmatory of numerous studies indicating mid-suture consists of stem and progenitor cells and osteogenesis occurs at the bone edge.

While we agree that the concept of stem cells in the suture and osteogenesis at the bone edges is relatively well established in mice, we feel Figs 1-4 contain a number of new contributions to the field. This is the first detailed cellular and molecular study of suture formation outside of mammals, and reinforces deep conservation of suture mechanisms across vertebrates. This is also, to our knowledge, the first detailed comparison of cell types before and after suture formation, revealing the presence of a similar cell population to suture mesenchyme at the leading edge of growing calvarial bones before suture establishment. Thus, this provides new insight into the embryonic origin of suture mesenchyme. We also provide in Fig 4 a new type of photoconversion-based lineage tracing for precisely quantifying osteoblast addition before and after suture formation in living animals. In general, we feel that establishing parallels between the cellular composition of sutures from fish to mammals is important for putting findings in the zebrafish system in a mammalian context, such as the mutant analysis we perform in Figs 7, 8, and S10.

The analysis of grem1a+ cells suggests that they are non-osteogenic, and do not contribute to bone formation. However, a report by Worthley et al has demonstrated that Grem1+ cells are SSCs contributing to bone formation in mice. There is no mention of this Cell paper published in 2015.

The authors need to discuss the discrepancy between their results and previous findings regarding the Grem1+ cell population.

Thank you for this important point. As mentioned above, we have now modified our text to emphasize that we cannot rule out that grem1a+ cells at cranial sutures also have a long-lived

SSC function. We now reference the Worthley et al. 2015 paper and discuss our observations in the context of this reporter and cranial suture SSC papers, as previously noted in Lines 406-427.

In Fig. 6C, did the authors perform photoconversion only on the boxed region? It was not clear how this experiment was performed.

This is correct and is in fact a distinct advantage of the nlsEOS lineage tracing approach where very specific subsets of expressing cells can be photoconverted and their subsequent lineages assessed. To clarify, we add the additional sentence to the figure legends:

Lines 665-666:

“*grem1a*+ cells were locally converted in the indicated boxed region and followed over up to 20 days post conversion (dpc).”

The authors should consider the co-localization study of *twist1b* and *tcf12* genes with RM markers, e.g., *grem1a*, *nog2*, *nog3*.

We have provided RNAscope for *twist1b* and the RM marker *angptl1b* (Fig. 7B) and confirm their co-expression within cranial sutures.

The reduction of *grem1a*+ cells in the fish craniosynostosis model *twist1b*; *tcf12* is a correlative observation. The results did not in any way indicate that *grem1a*+ cells have any regulatory function. The direct testing of *grem1a*+ cells on the regulation of osteogenic cells is missing but absolutely required for the conclusion. The authors need to perform functional studies of these cells. Due to the lack of direct evidence, the results have been overly interpreted, and the conclusion is premature at this stage.

As mentioned above, we have now performed analysis of triple zebrafish mutants lacking three Bmp antagonists enriched in the RM population (*grem1a*; *nog2*; *nog3*), with severe suture and calvarial bone phenotypes supporting a regulatory function of *grem1a*+ cells (new Fig. 8 and Fig.S10).

Reviewer #2 (Remarks to the Author):

The manuscript by Farmer et al. analyzes scRNAseq data from two different stages in zebrafish skull development, with a focus on defining cell populations at the forming sutures between bones of the calvarium. They distinguish two mesenchyme populations in the sutures: osteogenic, in close association with the edges of the bones; and regulatory, in the mid-sutures. Pseudotime analysis suggests a progression of cells from osteogenic mesenchyme to late osteoblasts, an interpretation supported by in situ localization of gene expression and by direct lineage tracing in both juvenile and adult fish.

Their findings contrast with prior mouse data suggesting that mesenchymal cells of the mid-suture contain osteoblast precursors, or that the underlying dura can give rise to new osteoblasts. However, they analyze previously acquired scRNAseq data sets from mouse sutures to show that many of the same cell types and clusters are present in the mouse, supporting conservation of suture organization and function between the two species. They localize expression in situ for

several fish orthologues of mouse markers, also supporting broad conservation of cell types. Finally, they propose a model where the mid-suture mesenchyme regulates the rate of bone growth and generation of new osteoblasts. Consistent with their model, they show that markers of the regulatory mesenchyme population are reduced or absent in a zebrafish model of Saethre-Chotzen syndrome, correlating with loss of the sutural gap and fusion of the bones.

Overall, the data are of excellent quality and support their conclusions. In particular, they have several lines of evidence supporting their main conclusion, that the osteoblast precursors reside near the edges of the growing skull bones. Without direct lineage tracing, they cannot conclude that the same is true in the mouse, but their data certainly suggest that it is. They also propose a model for the pathogenesis of Saethre-Chotzen syndrome, where the loss of regulatory mesenchyme is the underlying cause, rather than a cell-autonomous defect in the osteoblasts. They offer compelling data to support the model, although cell-autonomous effects could be key in other genetic craniosynostosis cases.

One minor criticism is that a few figure panels, currently showing single-color fluorescence, would be easier to see in greyscale (Fig 2D, E; Fig 3F; Fig 5C; Fig 6A; Fig 7B). Also, in the graphical abstract, the line colors of the osteogenic and regulatory mesenchyme are difficult to distinguish.

We thank the reviewer for the thoughtful response and recommendations. We have provided greyscale images for all single single-color fluorescent images (Fig 2D, E; Fig 3E; Fig 5C). We also changed the color scheme of the graphical abstract to better distinguish osteogenic and regulatory mesenchyme.

Reviewer #3 (Remarks to the Author):

In this study, the authors aimed to characterize suture cells in zebrafish using calvaria at juvenile bone front stages and adult stages by utilizing single-cell transcriptomics and integrating their findings with similar studies in mice. Through this approach, they identified conserved cell populations and validated them through in vivo expression analysis with several genetic tools. Furthermore, they discovered a novel non-osteogenic population of cells located in the mid-suture region after bones overlap, which they named regulatory mesenchyme (RM). While genetic analysis suggested that this population did not have osteogenic potential, the authors propose that it plays a role in restricting osteogenesis to sustain skull expansion. Although the study is comprehensive and the data are convincing, there are some issues with the identification and characterization of RM population that needs to be addressed.

Main comment 1

The authors named this non-osteogenic mesenchymal cell population regulatory mesenchyme (RM); however, there is currently no direct evidence demonstrating that this population indeed regulates osteogenesis. Although the authors showed that RM cells express genes related to BMP or TGF β signaling antagonists and provided insights into the relevance of this cell type to Saethre-Chotzen Syndrome through their experiments with *twist1b*; *tcf12* mutant zebrafish, a targeted cell ablation assay is necessary to fully support their claim.

We thank the reviewer for their thoughtful insight. While we currently lack the tools to specifically ablate regulatory mesenchyme, we now address a functional role of regulatory mesenchyme by generating three new zebrafish mutant lines for *grem1a*, *nog2*, and *nog3*, which represent the three main Bmp inhibitors that we found to be selectively enriched in this population. Although

individual mutants did not have significant phenotypes, we show that triple mutant combinations display accelerated bone growth, ectopic bone within the suture, and abnormal end-on-end suture morphology, consistent with a requirement for Bmp inhibitors from the *grem1a*⁺ population in regulating bone formation at the suture (new Fig. 8 and Fig S10). While this clearly supports RM-enriched Bmp antagonists in suture development, we also discuss why these new mutants only partially phenocopy craniosynostosis *twist1b*; *tcf12* mutants.

Lines 450-460:

“In both *twist1b*; *tcf12* mutants¹¹ and fish with severe losses of Bmp inhibitors expressed in the *grem1a*⁺ RM population (*grem1a*, *nog2*, *nog3*), we observed an initial acceleration of bone growth followed by a later stalling of bone growth, as well as ectopic bone formation. The failure of frontal and parietal bones to properly overlap at the coronal suture is also a common phenotype of *grem1a*^{-/-}; *nog2*^{-/-}; *nog3*^{+/-} mutant zebrafish and *Twist1*^{+/-}; *Tcf12*^{+/-} mutant mice. However, loss of Bmp antagonists does not phenocopy the coronal suture fusions seen in *twist1b*; *tcf12* mutants. This could be due to our inability to homozygose the *nog3* allele, redundancy with other Bmp antagonists expressed in the RM population (e.g. *fstl3*), or contribution of other RM genes to suture patency. For example, we note RM enrichment of *tgfb1* and *bambia*, which both negatively regulate the Tgf beta signaling pathway.”

Main comment 2

Related to the comment 1, performing a ligand-receptor analysis can provide additional insights into potential signaling interactions between the regulatory mesenchyme (RM) and osteoblast lineages. The authors need to address this assay. Additionally, the authors may consider validating the contributions of these signaling pathways in a bioassay. This would provide further evidence for the role of RM in regulating osteogenesis.

To address potential signaling interactions, we performed Gene Ontology (GO) analysis for each cluster and CellChat analysis to identify interactions between cell clusters (new Fig. S5). In support of RM identity, CellChat shows that the RM cluster has the second highest outgoing signaling score, making significant outgoing connections to osteogenic mesenchyme as predicted based on our model. We also find that osteoblasts have a high outgoing score and also connect to osteogenic mesenchyme, suggesting additional negative feedback to regulate osteoblast formation. While CellChat is not designed to detect inhibitory signaling, GO analysis identifies “blood vessel morphogenesis” and “regulation of Bmp signaling pathway” as top terms. In new data, we show that loss of the RM population in *twist1b*; *tcf12* mutants correlates with a loss of enriched blood vessels at the fused coronal sutures (Fig. 7G), and that loss of Bmp antagonists expressed in RM cells (*grem1a*; *nog2*; *nog3*) results in misregulated bone formation, ectopic bone in the suture, and abnormal end-on-end suture morphology. We provide Table S4 that highlights additional potential signaling interactions identified by CellChat that can be explored in future work.

Lines 182-195:

“To interrogate the transcriptional programs enriched within each cell type, we performed GO analysis using the significantly enriched genes for each cluster in our dataset (Fig. S5A). OM and osteoblast populations were enriched for terms associated with the skeletal system, cellular respiration, and extracellular matrix organization. RM displayed enrichment in blood vessel morphogenesis and regulation of Bmp signaling pathway, and meninges displayed enrichment for Bmp, Wnt, and retinoic acid signaling pathways. Periosteal dura is enriched for cartilage

development, consistent with the expression of cartilage-associated markers in its homologous population in the mouse coronal suture⁵. We next performed CellChat analysis²⁴ to assess potential signaling interactions between cell types. Whereas the OM and cartilage-related periosteal dura were predicted to have largely incoming signaling, the RM and late osteoblasts have largely outgoing signaling (**Fig. S5B**). Moreover, interaction mapping suggests the strongest predictive interactions arise from both RM and late osteoblasts toward periosteal dura and OM (**Fig. S5C, Table S4**). These data suggest that signaling from RM, and likely also negative feedback from differentiated osteoblasts, regulates OM and periosteal dura populations.”

Main comment 3

The study focuses on comparative analysis between zebrafish and mouse, but it is not clear whether the distinct gene expression signature of the RM population is conserved in both species. While the authors thoroughly characterized the RM population in zebrafish, they need to provide more information on the RM population in mouse, including its characterization in single-cell analysis and its spatial distribution in mouse calvaria suture. Addressing these points will be important for establishing the conservation of the RM population between species.

We thank the reviewer for this recommendation, as an RM population had not been previously reported in mice. We now provide a new Fig. S9 that formally tests the presence of an RM signature in mice. We re-evaluated our previously published mouse datasets for the expression of a set of the most specific RM markers. We identified two clusters, OG1 and OG3 which represent osteoprogenitors/suture mesenchyme and surface pre-osteoblasts/periosteum respectively, with the highest co-expression of RM markers. We also performed new in situ analysis in the mouse postnatal (P2) coronal sutures and confirmed enrichment of mouse homologs of zebrafish RM markers – *Nog*, *Tgbi*, *Angptl2* – in the mid-suture mesenchyme. This supports conservation of the RM program in the mouse cranial suture, although future lineage tracing will be needed to test this directly.

Main comment 4

Further computational analysis in the single cell RNA-seq data potentially sheds light on how the RM population emerges during skull development. By performing trajectory analysis and gene regulatory network analysis, the authors can identify the intermediate cell states and gene regulatory networks that give rise to the RM population. This will provide important insights into the developmental origin of the RM population and its role in regulating osteogenesis. The authors should consider performing such analyses and discussing its potential implications in their study.

We have now included RNA velocity to evaluate the possible lineage relationships between mesenchymal and osteogenic cells at cranial sutures. As described in response to Reviewer 1, this has been quite informative in showing strong connections from osteogenic mesenchyme to pre-osteoblasts, supporting our evidence for this being the primary osteogenic population. We also note some flow from regulatory mesenchyme to osteogenic mesenchyme. Based on this and prior data in mouse, we have now revised the text to more clearly state that the population we term regulatory mesenchyme may have functions both in restraining bone formation short-term and by providing a more long-term skeletal stem cell reservoir, though future long-term labeling strategies will be needed to test this directly. However, this analysis does not address the reviewer’s important point about the origins of the RM population. The RM population only differs in a few key genes (e.g. *grem1a*) from juvenile to adult stages and thus RM from different stages

do not cluster differently. We therefore could not make conclusions from RNA velocity about the emergence of the RM population, although we agree this is an important future direction.

Minor comment

I would recommend to omitting Fig. 3H, because current study does not fully support this scheme and more importantly, illustration of RM, a main finding in this study, is missing in the figure.

Thank you. We have removed 3H from the figure.

REVIEWER COMMENTS

Reviewer #1 (Remarks to the Author):

The revised manuscript “Cranial sutures spatially restrict osteogenesis to sustain skull expansion” has addressed some questions raised in the previous review. However, the most important concern remains unclear as to whether the identified *grem1a*⁺ cells are regulatory mesenchymal cells with true regulatory functions for osteogenic cells. Although the additional triple mutants of Bmp antagonists (*grem1a*, *nog2*, *nog3*) show cranial suture abnormalities. There is still no evidence supporting the regulatory function of *grem1a*⁺ cells. First, the triple mutants are global knockouts disrupting the genes in every cell. Although mainly expressed in the *grem1a*⁺ cell population, they are also present in other cell types evidenced by scRNA-seq analysis. Therefore, the authors cannot rule out the disruption of these antagonists in osteogenic cell types causing the observed skull defects. More importantly, the direct testing of *grem1a*⁺ cells on the regulation of osteogenic cells remains missing in this revision. These missing experiments and data are required to support their conclusion. It is also questionable as to no contribution of the *grem1a*⁺ cells to osteogenesis. Unfortunately, the provided cell tracking data lacks convincing evidence supporting the claims. The argument of short-term vs. long-term tracking seems illogical and unlikely to be the case. The interpretation of results from other published works is also inaccurately cited by the authors. Overall this is mainly a confirmatory study lacking rigor and critical functional data to support cranial sutures spatially restrict osteogenesis to sustain skull expansion.

Specific comments:

The triple mutants are global but not conditional knockouts. The *grem1a*, *nog2*, and *nog3* are expressed in the osteogenic mesenchyme in addition to the regulatory mesenchyme (Fig 5A). Because these genes are not specifically disrupted in the *grem1a*⁺ cells, their deletion in other cell types (e. g. skeletal stem/progenitor cells) may affect the osteogenic activity. Although the new data provide some support for the hypothesis, the function of *grem1a*⁺ cells in modulating skeletal stem/progenitor cells has yet to be rigorously tested.

The authors need to discuss the technical limitations of their genetic study and the possible functions of these genes in other cell types essential for regulating osteogenesis.

There is no evidence showing the loss of these antagonists altering BMP signaling in the mutants responsible for the observed suture and bone phenotypes.

The BMP signaling not only has been shown to stimulate bone formation but also stem cell regulation (Lines 430-434). A report by Maruyama et al indicated a regulatory role of *Bmpr1a* in SSCs and stemness. The deletion of *Bmpr1a* in the stem cells revealed bone island formation in the suture similar to the triple mutant described in this work. However, there is no mention of these key accomplishments.

The authors should also consider a discussion on BMP signaling in craniosynostosis based on their new genetic analyses of BMP antagonists.

There are reports isolating mouse and human SSCs from the suture mesenchyme with osteogenic ability to generate ectopic bones using transplantation analyses. The location of these SSCs had also been identified by specific stem cell markers better than the use of *Gli1* or *Prrx1* with a broad expression domain. Therefore, the previous results were beyond labeling a subset of cells using CreER and other techniques. While a higher resolution of the SSC profiling in the suture requires further investigation, the authors need to properly acknowledge prior contributions.

The *Gli1*⁺/*Axin2*⁺ cells contribute to bone formation after weeks not months of labeling (Zhao et al 2015, Maruyama et al 2016). The earliest time is 1 week to identify osteogenic cells derived from the stem cells in the suture mesenchyme. Therefore the revised statement (Lines 412-420) is incorrect.

The cell labeling analysis is cumulative over tracking time. Therefore, if the short-term tracking does not work, it is unlikely that the long-term tracking will show that *grem1a*⁺ cells have osteogenic potential. This appears to be a flaw.

How would the authors suggest there is still a long-term stem cell function of *grem1a*⁺ mid-suture cells? What is this statement based on? It is rather confusing and needs further clarification.

Line 141, Definition of long-term stem cells? Also, what is the short-term stem cells?

Cellular heterogeneity may play a role in the distinct functions of *grem1a*⁺ cells. However, it remains difficult to understand why there is no short-term labeling of osteogenic cells using *grem1a*⁺ tracking. How do the authors reconcile the triple mutants showing island bone formation in the suture at very early stages (11-13 mm SL) without the need for a long-term effect?

Fig 6, the authors need to show the efficiency of photoconversion in the suture area after UV light treatment. Also, there appear to be a few photoconverted cells not in the suture but embedded in the calvarial bones. The authors should show the neighboring bones in addition to the suture area in the same images. They also need to clearly define the boundary between the suture and bone.

In addition, do the triple mutants develop suture synostosis similar to the loss of *Bmpr1a* in the SSCs? Data provided in Fig 7E were not sufficient to show suture fusion. This needs to be demonstrated by histological evaluations and staining methods for ossification.

The authors need to provide quantitative analyses for the effect of the *twist1b:tcf12* mutations on blood vessels in Fig 7G. It is very difficult to see any difference between the two groups.

The authors still have not addressed this previously raised question: In Fig 3, the authors used pseudotime analysis with Monocle3 to identify the trajectory from osteogenic mesenchyme to late osteoblasts. However, it is not clear which root the authors select. How was the root determined?

Fig S10A, the *grem1a* mutants are not normal in comparison to the control. The suture looks wider and skull bones are smaller in the *grem1a* mutants.

The osteoblast in the metopic suture shows no significant alteration by the *grem1a* deletion (Fig S10D). However, the frontal bone growth is significantly reduced at 13-15 mm SL (Fig S10F). There is no explanation for such a discrepancy.

Inconsistency for the *nog2*^{+/-} in the text but *nog2*^{-/-} in the figure (Fig 8B-F).

Line 90, "...members of the Angiopoietin family linked to blood vessel development" needs citations.

Lines 441-445, this is a more definitive assessment of the regulatory function of the *grem1a*⁺ cell population. Unfortunately, these essential experiments are not performed to address the main questions asked in this manuscript.

Fig 7F, Fig S1C, no annotation for the color bars.

Fig S9, scale bars, not A-C.

Reviewer #2 (Remarks to the Author):

After the previous submission of this manuscript, I had only minor comments, which the authors have addressed. In addition, they have added substantial new data to support one of their main conclusions, that a major role of the mid-sutural regulatory mesenchyme is to modulate BMP signaling and thereby control bone growth. They show that mutation of multiple BMP inhibitors normally expressed by the RM cells results in aberrant suture formation and ectopic bone growth. In response to comments from the other reviewers, they have also added other data, including further analysis of their own scRNA data and corresponding mouse data sets, and expression analysis on mouse sutures. They have bent over backwards to satisfy the critiques, and as much as possible provide a unifying hypothesis to explain both mouse and zebrafish data.

Reviewer #3 (Remarks to the Author):

I accept the revised manuscript. The authors addressed all my comments and revised the manuscript properly.

Reviewer #1 (Remarks to the Author):

The revised manuscript “Cranial sutures spatially restrict osteogenesis to sustain skull expansion” has addressed some questions raised in the previous review. However, the most important concern remains unclear as to whether the identified *grem1a*⁺ cells are regulatory mesenchymal cells with true regulatory functions for osteogenic cells. Although the additional triple mutants of Bmp antagonists (*grem1a*, *nog2*, *nog3*) show cranial suture abnormalities. There is still no evidence supporting the regulatory function of *grem1a*⁺ cells. First, the triple mutants are global knockouts disrupting the genes in every cell. Although mainly expressed in the *grem1a*⁺ cell population, they are also present in other cell types evidenced by scRNA-seq analysis. Therefore, the authors cannot rule out the disruption of these antagonists in osteogenic cell types causing the observed skull defects. More importantly, the direct testing of *grem1a*⁺ cells on the regulation of osteogenic cells remains missing in this revision. These missing experiments and data are required to support their conclusion.

We add new data in this revision that BMP signaling is locally upregulated in the suture in the genetic absence of BMP antagonists enriched in the *grem1a*⁺ mid-suture mesenchyme. Combined with suture phenotypes upon loss of BMP antagonists and improved lineage tracing data showing that *grem1a*⁺ mid-suture cells do not contribute to osteoblasts, this further supports a regulatory role of *grem1a*⁺ mid-suture mesenchyme. However, we agree with the reviewer that this still needs to be definitively tested and have added text to the Discussion to reflect this. Unfortunately, Cre-Lox technology in zebrafish remains extremely difficult with only a handful of reports using this successfully. Creating tools to conditionally delete all 3 BMP antagonists would therefore take years of work, if successful at all. We also currently lack tools to ablate *grem1a*⁺ cells, and we feel that even if we could there are major caveats to interpreting phenotypes upon ablating the majority of cells in the mid-suture region.

To address the reviewer’s concerns, we have changed the title of the manuscript and now refer to *grem1a*⁺ mid-suture cells as “mid-suture mesenchyme” (MSM) rather than “regulatory mesenchyme” (RM). We feel the new title “BMP-dependent cellular dynamics during cranial suture establishment in zebrafish” now better reflects the substantial new insights into suture biology that were recognized by the other two reviewers. These include:

1. The first comprehensive single-analysis of suture development outside of mouse.
2. The first analysis of transcriptomic changes occurring during the transition from calvarial bone growth to suture establishment.
3. Generation of two new lineage tracing zebrafish lines that allow precise quantitation and spatial analysis of osteoblast formation at sutures and identification of a subset of mid-suture cells that are largely non-osteogenic.
4. Generation of three new Bmp antagonist mutants in zebrafish and demonstration of suture defects in triple mutant combinations.

We have also added text to the Discussion recognizing the important caveats pointed out by the reviewer.

Lines 433-437: “While OM markers are enriched at the suture edges where the majority of new osteoblasts form, we also detected OM marker expression in a few cells within the mid-suture region. Our findings are therefore consistent with heterogeneity of mid-suture mesenchyme, with *grem1a*⁺ cells in this domain potentially regulating the behavior of neighboring skeletal stem cells both within the mid-suture region and suture edges.”

Lines 452-457: “However, future studies that specifically remove these antagonists within suture mesenchyme will be needed to definitively demonstrate requirements for BMP antagonists in the MSM population for suture regulation. It will also be interesting to assess the effect of ablating the *grem1a*⁺ mesenchyme population on suture formation and maintenance, as well as identifying the upstream signals that activate this MSM expression program during suture establishment.”

It is also questionable as to no contribution of the *grem1a*⁺ cells to osteogenesis. Unfortunately, the provided cell tracking data lacks convincing evidence supporting the claims. The argument of short-term vs. long-term tracking seems illogical and unlikely to be the case.

We now provide additional data in Fig 6 that shows more convincingly that *grem1a*:nlEOS cells do not contribute to osteoblasts. Rather than describing tracking in terms of “short term” or “long term”, we now more precisely state that our data indicate no contributions within one month. In Fig. 6C, we provide clearer images that include more of the frontal and parietal bones and show that photoconverted *grem1a*:nlEOS cells remain in the suture and do not contribute to bone within 28 days. These data also demonstrate that *grem1a*:nlEOS protein survives for at least 28 days after it is produced. In order to more directly show lack of contribution of *grem1a*:nlEOS cells to osteoblasts, we now perform co-localization of *grem1a*:nlEOS with the osteoblast marker *sp7*:mCherry in new Fig. 6D. We then use quantitation in Fig. 6E to show that virtually no *grem1a*:nlEOS cells turn on the *sp7*:mCherry osteoblast marker.

Lines 312-325: “To test if nlEOS can function as a lineage reporter, we tracked photoconverted skulls one month after UV treatment (**Fig. 6B, C**). Previous observations suggest that nlEOS protein can perdure for several weeks after expression of nlEOS mRNA ceases^{33,34}, and we find that mid-suture mesenchyme retains detectable converted protein and accumulates new nlEOS protein, dynamics consistent with its use as a lineage reporter (**Fig. 6B, C**). During this time period, *grem1a*:nlEOS⁺ cells remained confined to the mid-suture region and did not noticeably change their distribution. Despite nlEOS protein stability over several weeks, we observed no co-expression of nlEOS with the osteoblast marker *sp7*:mCherry, showing that *grem1a*⁺ cells do not generate osteoblasts within a month (**Fig. 6D, E**). In contrast, quantification of de novo osteoblast at adult stages (>24 mm) using *osteoblast*:nlEOS confirmed sustained osteogenesis at all cranial sutures within a two-week period (**Fig. S10A-C**). These results point to *grem1a*⁺ MSM cells not being a major contributor to new osteoblasts, although we cannot rule out that they make contributions to osteoblasts in non-homeostatic conditions.”

Specific comments:

The triple mutants are global but not conditional knockouts. The *grem1a*, *nog2*, and *nog3* are expressed in the osteogenic mesenchyme in addition to the regulatory mesenchyme (Fig 5A). Because these genes are not specifically disrupted in the *grem1a*⁺ cells, their deletion in other cell types (e. g. skeletal stem/progenitor cells) may affect the osteogenic activity. Although the new data provide some support for the hypothesis, the function of *grem1a*⁺ cells in modulating skeletal stem/progenitor cells has yet to be rigorously tested.

The authors need to discuss the technical limitations of their genetic study and the possible functions of these genes in other cell types essential for regulating osteogenesis.

We now discuss this important caveat in the Discussion.

Lines 452-455: “However, future studies that specifically remove these antagonists within suture mesenchyme will be needed to definitively demonstrate requirements for BMP antagonists in the MSM population for suture regulation.”

There is no evidence showing the loss of these antagonists altering BMP signaling in the mutants responsible for the observed suture and bone phenotypes.

In new Fig. 8K,L, we now show that the proportion of suture cells positive for phosphorylated SMAD1/5, a marker of active Bmp signaling, is increased in combinatorial BMP antagonist mutants.

The BMP signaling not only has been shown to stimulate bone formation but also stem cell regulation (Lines 430-434). A report by Maruyama et al indicated a regulatory role of *Bmpr1a* in SSCs and stemness. The deletion of *Bmpr1a* in the stem cells revealed bone island formation in the suture similar to the triple mutant described in this work. However, there is no mention of these key accomplishments.

The authors should also consider a discussion on BMP signaling in craniosynostosis based on their new genetic analyses of BMP antagonists.

We thank the authors for this recommendation and have expanded our discussion to include this paper and additional papers that demonstrate requirements for BMP signaling at cranial sutures:

Line 444-448: “Loss of *Bmpr1a* in *Axin2*⁺ skeletal stem cells impairs self-renewal and leads to ectopic bone formation and craniosynostosis⁴¹, and expression of constitutive active *Bmpr1a* in neural crest lineage cells leads to premature fusion of the metopic suture⁴². These data demonstrate a critical need for tight control of BMP signaling at cranial sutures.”

There are reports isolating mouse and human SSCs from the suture mesenchyme with osteogenic ability to generate ectopic bones using transplantation analyses. The location of these SCCs had also been identified by specific stem cell markers better than the use of *Gli1* or *Prrx1* with a broad expression domain. Therefore, the previous results were beyond labeling a subset of cells using CreER and other techniques. While a higher resolution of the SSC profiling in the suture requires further investigation, the authors need to properly acknowledge prior contributions.

We now include references to reports that use FACS isolation to identify stem cell populations at cranial sutures as recommended by the reviewer.

Lines 64-65: “Further, suture-residing skeletal stem cells in mice and humans have been captured using surface antigens associated with stem cell identity^{9,11,12}.”

The *Gli1*⁺/*Axin2*⁺ cells contribute to bone formation after weeks not months of labeling (Zhao et al 2015, Maruyama et al 2016). The earliest time is 1 week to identify osteogenic cells derived

from the stem cells in the suture mesenchyme. Therefore the revised statement (Lines 412-420) is incorrect.

We apologize for the oversight. We have removed discussion of these stages as they were discussed in the context of proposing a *grem1a*⁺ cells as long-lived stem cells, which we now removed.

The cell labeling analysis is cumulative over tracking time. Therefore, if the short-term tracking does not work, it is unlikely that the long-term tracking will show that *grem1a*⁺ cells have osteogenic potential. This appears to be a flaw. How would the authors suggest there is still a long-term stem cell function of *grem1a*⁺ mid-suture cells? What is this statement based on? It is rather confusing and needs further clarification. Also, what is the short-term stem cells?

We agree that this was confusing and as stated above no longer refer to stem cell as short or long term. Instead, we more precisely state our findings that we see no contribution of *grem1a*:nl^sEOS cells to osteoblasts over the course of one month.

Cellular heterogeneity may play a role in the distinct functions of *grem1a*⁺ cells. However, it remains difficult to understand why there is no short-term labeling of osteogenic cells using *grem1a*⁺ tracking. How do the authors reconcile the triple mutants showing island bone formation in the suture at very early stages (11-13 mm SL) without the need for a long-term effect?

We now discuss cellular heterogeneity at the suture and more clearly explain that our data support a role for *grem1a*⁺ cells in signaling to nearby bone-forming cells in the suture.

Lines 430-437: "In contrast to the non-osteogenic properties of *grem1a*⁺ MSM, pseudotime analysis points to *hyla4*⁺/*fli1a*⁺ OM as a likely skeletal stem cell population, although future lineage tracing will be required to confirm this. While OM markers are enriched at the suture edges where the majority of new osteoblasts form, we also detected OM marker expression in a few cells within the mid-suture region. Our findings are therefore consistent with heterogeneity of mid-suture mesenchyme, with *grem1a*⁺ cells in this domain potentially regulating the behavior of neighboring skeletal stem cells both within the mid-suture region and suture edges."

Fig 6, the authors need to show the efficiency of photoconversion in the suture area after UV light treatment. Also, there appear to be a few photoconverted cells not in the suture but embedded in the calvarial bones. The authors should show the neighboring bones in addition to the suture area in the same images. They also need to clearly define the boundary between the suture and bone.

In revised Fig. 6C, we show that photoconversion in the suture area is nearly 100% efficient (near complete absence of unconverted green signal). We have also expanded the view to visualize neighboring parietal and frontal bones. The few speckles of signal outside the suture are in the skin and we now confirm in Fig. 6D,E that virtually no *grem1a*:nl^sEOS cells express the *sp7*:mCherry osteoblast marker.

In addition, do the triple mutants develop suture synostosis similar to the loss of *Bmpr1a* in the SSCs?

While we observe ectopic bone and defects in suture morphology, triple mutants do not develop synostosis. We now discuss several explanations for why sutures do not fuse in these mutants.

Lines 471-476: “However, we failed to observe the coronal suture fusions of *twist1b*; *tcf12* mutants in animals with combinatorial BMP antagonist loss. This could be due to the juvenile lethality of these mutants, our inability to homozygose the *nog3* allele, redundancy with other BMP antagonists expressed in the MSM population (e.g. *fstl3*), and/or contribution of other MSM genes to suture patency. For example, we note MSM enrichment of *tgfb1* and *bambia*, which both negatively regulate the Tgf beta signaling pathway.”

Data provided in Fig 7E were not sufficient to show suture fusion. This needs to be demonstrated by histological evaluations and staining methods for ossification.

We have previously reported an extensive characterization of suture fusions in *twist1b*; *tcf12* mutants, including histology and direct bone staining (Teng et al, 2018). In the text, we now better explain how appearance of new (green) *osteoblast:nlsEOS* osteoblasts and increased signal intensity due to overlapping bones at the sutures are good proxies of suture patency.

Line 334-339: “Our *osteoblast:nlsEOS* reporter allows to us to quantify osteoblast differentiation and readily visualize the location of cranial sutures by detection of de novo osteoblasts (green) and increased signal from two overlapping bones covered by osteoblasts. Using *osteoblast:nlsEOS* to quantify osteoblast addition, we found that loss of *grem1a:nlsEOS*+ MSM cells in *twist1b*^{-/-}; *tcf12*^{-/-} mutants correlated with increased osteoblast formation at growing bone fronts (Fig. 7E,F).”

The authors need to provide quantitative analyses for the effect of the *twist1b*:*tcf12* mutations on blood vessels in Fig 7G. It is very difficult to see any difference between the two groups.

In Fig. 7G,H we now provide clearer images and quantification of reduced vessel density at sutures of *twist1b*; *tcf12* mutants.

The authors still have not addressed this previously raised question: In Fig 3, the authors used pseudotime analysis with Monocle3 to identify the trajectory from osteogenic mesenchyme to late osteoblasts. However, it is not clear which root the authors select. How was the root determined?

We now describe the rationalization for root selection.

Lines 231-234: “We used OM as the root based on its shared gene expression program with osteoprogenitors from mouse and zebrafish scRNA-seq datasets^{5,20}, and the transcriptional flow from OM to committed osteoblast identities in our RNA velocity analysis.”

Fig S10A, the *grem1a* mutants are not normal in comparison to the control. The suture looks wider and skull bones are smaller in the *grem1a* mutants.

We apologize for this oversight as these images did not reflect size-matched animals. We also now acknowledge that skull morphology was not comprehensively examined.

Lines 363-364: “While skull shape was not obviously abnormal in *grem1a* mutants, we did not perform a comprehensive analysis.”

The osteoblast in the metopic suture shows no significant alteration by the *grem1a* deletion (Fig S10D). However, the frontal bone growth is significantly reduced at 13-15 mm SL (Fig S10F). There is no explanation for such a discrepancy.

We now provide a rationalization for the observed differences.

Lines 358-362: “We observed a significant deceleration of frontal bone growth in *grem1a* mutants between 13 and 15 mm SL stages by serially staining with different color bone dyes as previously reported¹³, which may reflect a higher sensitivity of the assay to capture cumulative bone growth changes compared to quantification of de novo osteoblasts during the same growth period (Fig. S10E-G).”

Inconsistency for the *nog2*^{+/-} in the text but *nog2*^{-/-} in the figure (Fig 8B-F).

Thank for this observation. We have made the appropriate changes throughout the manuscript.

Line 90, “...members of the Angiopoietin family linked to blood vessel development” needs citations.

We have added relevant citations.

Lines 441-445, this is a more definitive assessment of the regulatory function of the *grem1a*⁺ cell population. Unfortunately, these essential experiments are not performed to address the main questions asked in this manuscript.

We now address this caveat in the Discussion as stated above.

Fig 7F, Fig S1C, no annotation for the color bars.

We changed from color to greyscale in Fig. 7F and added the appropriate color bar for Fig. S1C.

Fig S9, scale bars, not A-C.

We have corrected this.

REVIEWERS' COMMENTS

Reviewer #1 (Remarks to the Author):

This is a second revision of the manuscript, now with a new title “BMP-dependent cellular dynamics during cranial suture establishment in zebrafish”. The strengths include experiments examining cranial sutures using single cell transcriptomics, cell tracing analysis, and genetically modified fish models. The authors mainly addressed criticisms raised in the previous review by backing down their statements and withdrawing their conclusions that *grem1a+* cells are regulatory mesenchymal cells for osteogenic cells. The significance of this work is thus reduced greatly. In addition, a few key issues remain outstanding – apparent weaknesses. First, no contribution of the *grem1a+* cells to osteogenesis in zebrafish that is contrary to the mouse study (Worthley et al., Cell 2015) is still puzzling. The authors present inconsistent results with one showing non osteogenic and another implying latent osteogenesis of the *grem1a+* cell population. There have been increasing notions about the heavy dependence of cell tracing analysis for studying cranial sutures. This is because only a handful of reports have made extensive efforts to obtain compelling evidence for functional characterizations of the identified cell populations, e.g. osteogenic and regenerative potentials of the identified stem cells. Second, the lack of rigorous data supporting the function of *grem1a+* cells is in line with the emerging concern in the craniofacial biology field. These concerns are further manifested by the inability to reconcile the triple mutants showing island bone formation in the suture at very early stages (11-13 mm SL) without needing a long-term/latent effect. The authors’ speculations seem to contradict each other. The authors suggested that *grem1a+* MSM (mid-suture mesenchyme) has minimal osteogenesis effect. However, RNA velocity showed a directional flow from *grem1a+* MSM towards OM (osteogenic mesenchyme), indicating a possible skeletal stem/progenitor cell population. Finally, the direct link between the function of *grem1a+* cells and the ectopic bone formation caused by the loss of MSM-enriched BMP antagonists is still missing. The *grem1a+* tracking in the triple mutants may provide useful information but is not included in the current study. Functional analyses, primarily correlative observations, are the weakest of all. Without convincing evidence, the outcomes are inconclusive and speculative. It’s disappointing that the authors did not explore new ideas and could not uncover novel molecular or cellular regulation using single cell transcriptomics. The role of BMP antagonists in suture patency has been known for more than 20 years (Warren et al, Nature 2003). In summary, the revised manuscript remains mainly confirmatory of existing knowledge and lacks novelty, resulting in incremental advancements.

This is a second revision of the manuscript, now with a new title “BMP-dependent cellular dynamics during cranial suture establishment in zebrafish”. The strengths include experiments examining cranial sutures using single cell transcriptomics, cell tracing analysis, and genetically modified fish models. The authors mainly addressed criticisms raised in the previous review by backing down their statements and withdrawing their conclusions that *grem1a*⁺ cells are regulatory mesenchymal cells for osteogenic cells. The significance of this work is thus reduced greatly.

We believe our data still strongly support a role of *grem1a*⁺ suture cells in having a non-osteogenic regulatory function. Our single-cell analysis shows that *grem1a* and several other Bmp antagonists are uniquely expressed in a suture-specific mesenchymal population. Lineage tracing with a *grem1a:nlsEOS* line shows that these suture-specific mesenchymal cells do not contribute to osteoblasts over several weeks of development. In addition, genetic loss of *grem1a* along with Bmp antagonists *nog2* and *nog3* disrupts suture formation. We have therefore not withdrawn our conclusions that *grem1a*⁺ cells are regulatory mesenchymal cells, but rather acknowledge that future experiments, such as conditional loss of Bmp antagonists in *grem1a*⁺ suture cells, will be needed to definitively prove our model.

In addition, a few key issues remain outstanding – apparent weaknesses. First, no contribution of the *grem1a*⁺ cells to osteogenesis in zebrafish that is contrary to the mouse study (Worthley et al., Cell 2015) is still puzzling. The authors present inconsistent results with one showing non osteogenic and another implying latent osteogenesis of the *grem1a*⁺ cell population. There have been increasing notions about the heavy dependence of cell tracing analysis for studying cranial sutures. This is because only a handful of reports have made extensive efforts to obtain compelling evidence for functional characterizations of the identified cell populations, e.g. osteogenic and regenerative potentials of the identified stem cells.

The Worthley et al. study investigated Grem1 as a marker of skeletal stem cells in long bones. It is therefore plausible that Grem1⁺ cells from the Worthley study represent a unique stem cell population that supports endochondral bone growth, in contrast to the stem cells supporting intramembranous bone growth in the calvarial bones separated by sutures. Our unique approach using imaging-based lineage tracing of *grem1a:nlsEOS*⁺ suture cells clearly shows no contributions to bone over several weeks of development. However, given the limitations of the nlsEOS-based lineage tracing approach, we cannot rule out that *grem1a*⁺ suture cells may contribute to bone over much longer timescales.

Second, the lack of rigorous data supporting the function of *grem1a*⁺ cells is in line with the emerging concern in the craniofacial biology field. These concerns are further manifested by the inability to reconcile the triple mutants showing island bone formation in the suture at very early stages (11-13 mm SL) without needing a long-term/latent effect. The authors’ speculations seem to contradict each other. The authors suggested that *grem1a*⁺ MSM (mid-suture mesenchyme) has minimal osteogenesis effect. However, RNA velocity showed a directional flow from *grem1a*⁺ MSM towards OM (osteogenic mesenchyme), indicating a possible skeletal stem/progenitor cell population. Finally, the direct link between the function of *grem1a*⁺ cells and the ectopic bone formation caused by the loss of MSM-enriched BMP antagonists is still missing. The *grem1a*⁺ tracking in the triple mutants may provide useful information but is not included in the current study. Functional analyses, primarily correlative observations, are the weakest of all. Without convincing evidence, the outcomes are inconclusive and speculative. It’s disappointing that the authors did not explore new ideas and could not uncover novel molecular or cellular regulation using single cell transcriptomics.

Based on single-cell RNA sequencing, in situ hybridization, and analysis of a newly generated *grem1a:nlsEOS* transgenic line, we clearly observe upregulation of *grem1a* and other Bmp antagonists in sutures at the earliest stages of their formation, in line with ectopic bone formation in mutants at 11-13 mm SL. We do not propose a long-term/latent effect. While RNA velocity suggested that some *grem1a*⁺ MSM may flow towards osteogenic mesenchyme, our *grem1a:nlsEOS* lineage tracing data do not support this in the short term (several weeks). We believe that in vivo functional testing is more definitive than bioinformatics predictions, though, as stated above, we cannot rule out that *grem1a*⁺ cells contribute to osteoblasts beyond stages at which nlsEOS can be used for lineage tracing. While we agree that lineage tracing of *grem1a:nlsEOS*⁺ cells in the triple mutant will be

informative in the future, the number of alleles required to construct these fish was beyond the timeframe of this revision cycle.

The role of BMP antagonists in suture patency has been known for more than 20 years (Warren et al, Nature 2003).

The Warren et al. paper demonstrates that overexpression of the Bmp antagonist Noggin is sufficient to prevent suture fusion, but does not test the functional requirement of Bmp antagonists at cranial sutures. We acknowledge this paper and believe our results highlight the function of multiple BMP antagonists, including Noggin, at cranial sutures. To our knowledge, our study is the first to demonstrate a requirement for Bmp antagonists for the timely establishment of cranial sutures. These results highlight the complex function of Bmp signaling at cranial sutures, and a need to further explore cell-type specific functions of the pathway during calvaria development.

In summary, the revised manuscript remains mainly confirmatory of existing knowledge and lacks novelty, resulting in incremental advancements.

We disagree with the reviewer and believe that this study makes significant new contributions to our understanding of cranial suture formation, as outlined below:

1. We provide the most comprehensive single-cell transcriptomic analysis of suture development outside of mouse. By identifying common suture cell populations between zebrafish and mouse, we reveal deeply conserved cell and molecular processes underlying suture establishment in all vertebrates.
2. Our study is the first to identify transcriptomic changes occurring during the transition from calvarial bone growth to suture establishment. In particular, we reveal that upregulation of Bmp antagonist and pro-angiogenic gene expression during suture establishment helps explain the slowing of Bmp-dependent bone growth and increased vascularization within sutures.
3. We generate two new lineage tracing zebrafish lines that allow precise quantitation and spatial analysis of osteoblast formation at sutures, as well as identification of a novel subset of mid-suture cells that have a regulatory and non-osteogenic role in controlling bone addition.
4. By generating three new Bmp antagonist mutants in zebrafish and breeding triple mutants, we are the first to show genetic requirements of Bmp antagonists in suture establishment.